# A Fast Scale-Invariant Algorithm for Non-negative Least Squares with Non-negative Data*

**Jelena Diakonikolas**
University of Wisconsin-Madison
jelena@cs.wisc.edu

**Chenghui Li**
University of Wisconsin-Madison
cli539@wisc.edu

**Swati Padmanabhan**
University of Washington, Seattle
pswati@uw.edu

**Chaobing Song**
University of Wisconsin-Madison
chaobing.song@wisc.edu

## Abstract

Nonnegative (linear) least square problems are a fundamental class of problems that is well-studied in statistical learning and for which solvers have been implemented in many of the standard programming languages used within the machine learning community. The existing off-the-shelf solvers view the non-negativity constraint in these problems as an obstacle and, compared to unconstrained least squares, perform additional effort to address it. However, in many of the typical applications, the data itself is nonnegative as well, and we show that the nonnegativity in this case makes the problem easier. In particular, while the worst-case dimension-independent oracle complexity for unconstrained least squares problems necessarily scales with one of the data matrix constants (typically the spectral norm) and these problems are solved to additive error, we show that nonnegative least squares problems with nonnegative data are solvable to multiplicative error and with complexity independent of any matrix constants. The algorithm we introduce is accelerated and based on a primal-dual perspective. We further show how to provably obtain linear convergence using adaptive restart coupled with our method and demonstrate its effectiveness on large-scale data via numerical experiments.

## 1 Introduction

Nonnegative least squares (NNLS) problems, defined by $\min_{\mathbf{x} \geq \mathbf{0}} \quad \frac{1}{2}\|\mathbf{A}\mathbf{x} - \mathbf{b}\|_2^2$, where $\mathbf{A} \in \mathbb{R}^{m \times n}$ and $\mathbf{b} \in \mathbb{R}^m$, are fundamental problems and have been studied for decades in optimization and statistical learning [26, 7, 23], with various off-the-shelf solvers available in standard packages of Python (as `optimize.nnls` in the SciPy package), Julia (as `nnls.jl`), and MATLAB (as `lsqnonneg`). Within machine learning, NNLS problems arise whenever having negative labels is not meaningful, for example, when representing prices, age, pixel intensities, chemical concentrations, or frequency counts. NNLS is also widely used as a subroutine in nonnegative matrix factorization [10, 18, 25] to extract sparse features in applications like clustering, collaborative filtering, and community detection.

From a statistical perspective, NNLS problems can be shown to possess a regularization property that enforces sparsity similar to LASSO [46], while being comparatively simpler, without the need to tune a regularization parameter or perform cross-validation [44, 8, 17, 24, 49, 43].

From an algorithmic standpoint, the nonnegativity constraint in NNLS problems is typically viewed as an obstacle: most NNLS algorithms perform additional work to handle it, and the problem is

---

*Authors ordered alphabetically.

36th Conference on Neural Information Processing Systems (NeurIPS 2022).

considered harder than unconstrained least squares. However, in many important applications of NNLS, such as text mining [6], functional MRI [3, 20], EEG data analysis [33], pulse oximetry [22, 50], statistical procedures in observational astronomy [19], and those traditionally addressed using nonnegative matrix factorization [10], *the data is also nonnegative*. We argue in this paper that when the data for NNLS is nonnegative, it is in fact possible to obtain *stronger* guarantees than for traditional least squares.

**Our Contributions.** We study NNLS problems with (element-wise) nonnegative data matrix $\mathbf{A}$, to which we refer as the NNLS+ problems, through the lens of the (equivalent) quadratic problems:

$$\min_{\mathbf{x} \geq \mathbf{0}} \left\{ \bar{f}(\mathbf{x}) \stackrel{\text{def}}{=} \frac{1}{2} \|\mathbf{A}\mathbf{x}\|_2^2 - \mathbf{c}^\top \mathbf{x} \right\}, \tag{P}$$

where $\mathbf{c} = \mathbf{A}^\top \mathbf{b}$ may be assumed element-wise positive. This assumption is without loss of generality since if $c_j < 0$ for some $j$, then $\nabla_j \bar{f}(\mathbf{x}) \geq 0$, implying that the $j^{\text{th}}$ coordinate of the optimal solution is zero[2]. Hence, we could fix $x_j = 0$ and optimize over only the remaining coordinates.

We further assume that the matrix $\mathbf{A}$ is non-degenerate: none of its rows or columns has all elements equal to zero. This assumption is without loss of generality because (1) if such a row exists, we could remove it without affecting the objective, and (2) if the $j^{\text{th}}$ column had all elements equal to zero, the optimal value of (P) would be $-\infty$, obtained for $\mathbf{x}$ with $x_j \to \infty$. Having established our assumptions and setup, we now proceed to state our contributions, which are three-fold.

**(1) A scale-invariant, $\varepsilon$-multiplicative algorithm.** We design an algorithm based on coordinate descent that, in total cost $O(\frac{\text{nnz}(\mathbf{A})}{\sqrt{\varepsilon}})$, constructs an $\varepsilon$-*multiplicative* approximate solution to (P). Our algorithm capitalizes on structural properties of (P) that arise as a result of the nonnegativity of $\mathbf{A}$.

**Theorem 1.1** (Informal; see Theorem 3.5). *Given a matrix $\mathbf{A} \in \mathbb{R}_+^{m \times n}$ and $\varepsilon > 0$, define $f(\mathbf{x}) = \frac{1}{2} \|\mathbf{A}\mathbf{x}\|_2^2 - \mathbf{c}^\top \mathbf{x}$ and $\mathbf{x}^\star \in \operatorname{argmin}_{\mathbf{x} \geq \mathbf{0}} f(\mathbf{x})$. Then, there exists an algorithm that in $K = O(n \log n + \frac{n}{\sqrt{\varepsilon}})$ iterations and $O\big(\text{nnz}(\mathbf{A})\big(\log n + \frac{1}{\sqrt{\varepsilon}}\big)\big)$ arithmetic operations returns $\widetilde{\mathbf{x}}_K \in \mathbb{R}_+^n$ such that $\mathbb{E}\left[\langle \nabla f(\widetilde{\mathbf{x}}_K), \widetilde{\mathbf{x}}_K - \mathbf{x}^\star \rangle\right] \leq \varepsilon |f(\mathbf{x}^\star)|$.*

The application of our structural observations on (P) to Theorem 4.6 of [13] enables the recovery of our guarantee on the optimality gap; however, we provide a guarantee on the primal-dual gap, and this is *stronger* than the one on the optimality gap stated in Theorem 1.1. What is significant about Theorem 1.1 is the *invariance* of the computational complexity to the scale of $\mathbf{A}$—it does not depend on any matrix constants. This cost stands in stark contrast to that of traditional least squares, where the dependence of (oracle) complexity on matrix constants (specifically, the spectral norm of $\mathbf{A}$ in the Euclidean case) is *unavoidable* [35, 36], and multiplicative approximation is *not possible in general*.[3] In general, scale-invariance is a crucial feature in problems with data matrices, since a dependence on the width implies that the algorithm is technically not polynomial-time. This feature has, in fact, been an object of extensive study in the long line of works on packing and covering linear programs [48, 1] and its variants such as a fair packing [14]. Conceptually, our algorithm is a new acceleration technique inspired by VRPDA[2] [45].

**(2) Linear convergence with restart.** By incorporating adaptive restart in (P), we improve the guarantee of Theorem 1.1 to one with linear convergence (with $\log(1/\varepsilon)$ complexity). Thus, we establish the first theoretical guarantee for NNLS+ that simultaneously satisfies the properties of being *scale-invariant, accelerated, and linearly-convergent*.

**Theorem 1.2** (Informal; see Theorem 4.1). *Consider the setup of Theorem 1.1. Then, there is an algorithm that in expected $O(\text{nnz}(\mathbf{A})(\log n + \frac{\sqrt{n}}{\mu}) \log(\frac{1}{\varepsilon}))$ arithmetic operations returns $\widetilde{\mathbf{x}}_K \in \mathbb{R}_+^n$ with $f(\widetilde{\mathbf{x}}_K) - f(\mathbf{x}^\star) \leq \varepsilon |f(\mathbf{x}^\star)|$, where $\mu$ is the constant in a local error bound for (P).*

Proving this bound requires bounding the expected number of iterations between restarts in conjunction with careful technical work in identifying an appropriate local error bound for NNLS+.

---

[2]To see this, note that by first-order optimality condition $\nabla \bar{f}(\mathbf{x}^\star)^\top (\mathbf{x} - \mathbf{x}^\star) \geq 0$ for all $\mathbf{x} \geq \mathbf{0}$. Choosing $\mathbf{x}$ with $\mathbf{x}_i = \mathbf{x}_i^\star$ for all $i \neq j$ and $x_j = 0$ in the first-order optimality condition gives $\mathbf{x}_j^\star = 0$.

[3]To see why, consider a case in which the optimal objective value equals zero. Then any problem with a multiplicative guarantee of the form in Theorem 1.1 would necessarily return an optimal solution.

**(3) Numerical experiments.** We consolidate our theoretical contributions with a demonstration of the empirical advantage of our restarted method over state-of-the-art solvers via numerical experiments on datasets from LibSVM with sizes up to $19996 \times 1355191$. Figure 1 shows that, when combined with the restart strategy, our algorithm significantly outperforms the compared algorithms.

**Related work.** NNLS has seen a large body of work on the empirical front. The first method that was widely adopted in practice (including in the `lsqnonneg` implementation of MATLAB) is due to the seminal work of [26] (originally published in 1974). This method, based on active sets, solves NNLS via a sequence of (unconstrained) least squares problems and has been followed up by [7, 47, 34, 11] with improved empirical performance. While these variants are generally effective on small to mid-scale problem instances, they are not suitable for extreme-scale problems ubiquitous in machine learning. For example, in the experiments reported in [34], Fast NNLS [7] took 6.63 days to solve a problem of size $45000 \times 45000$, while the TNT-NN algorithm [34] took 2.45 hours. However, the latter requires computing the Cholesky decomposition of $\mathbf{A}^\top \mathbf{A}$ at initialization, which can be prohibitively expensive both in computation and in memory. Another prominent work on the empirical front is that of [23], which performs projected gradient descent with modified Barzilai-Borwein steps [4] and step sizes a carefully designed sequence of diminishing scalars.

Another, separate, line of work concerns optimization algorithms with multiplicative error guarantees. Of interest to us are standard first-order algorithms that run in time that is near-linear in the input size and are thus applicable to large-scale setting (for multiplicative-error algorithms applicable to broad classes of problems but that run in time that is superlinear in the input size, see [41, Chapter 7]). Most of the existing literature in this domain concerns positive (packing and covering) linear programs (e.g., [29, 51, 1, 30]). Results also exist for positive semidefinite programs [2], (nonlinear) fair packing and covering problems [32, 14], and fair packing problems under Schatten norms for matrices [21]. With the exception of [14] (discussed below), none of these results are directly applicable to (P).

To the best of our knowledge, theoretical guarantees explicitly for (P) have been scarce. For instance, [23] mentioned in the preceding paragraph provides only asymptotic guarantees. Orthogonally, the result on 1-fair covering by [14] solves the dual of NNLS+, which also gives a multiplicative guarantee for NNLS+, but with overall complexity $\tilde{O}\left(\frac{\mathrm{nnz}(\mathbf{A})}{\varepsilon}\right)$, where $\tilde{O}(\cdot)$ hides poly-log factors.

Since our algorithm is based on the coordinate descent algorithm, we highlight some results of other coordinate descent algorithms when specialized to the closely related problem of *unconstrained linear regression*. The pioneering work of [37] proposed a coordinate descent method called `RCDM`, which in our setting has an iteration cost $O\left(\frac{\sum_{j=1}^{n} \|\mathbf{A}_{:j}\|_2^2 \|\mathbf{x}_0 - \mathbf{x}^\star\|^2}{\varepsilon}\right)$, where $\|\mathbf{A}_{:j}\|_2$ is the Euclidean norm of the $j^{\text{th}}$ column of $\mathbf{A}$. This was improved by [27], in an algorithm termed `ACDM`, by combining Nesterov's estimation technique [39] and coordinate sampling, giving an iteration complexity of $O\left(\frac{\sqrt{n \sum_{j=1}^{n} \|\mathbf{A}_{:j}\|_2^2} \|\mathbf{x}_0 - \mathbf{x}^\star\|}{\sqrt{\varepsilon}}\right)$ for solving (P). The latest results in this line of work by [2, 42, 40] perform non-uniform sampling atop a framework of [37] and achieve iteration complexity of $O\left(\frac{\sqrt{\sum_{j=1}^{n} \|\mathbf{A}_{:j}\|_2^2} \|\mathbf{x}_0 - \mathbf{x}^\star\|}{\sqrt{\varepsilon}}\right)$, with [12] dropping the dependence on $\max_{1 \le j \le n} \|\mathbf{A}_{:j}\|_2$. Additionally, the work of [28] develops an accelerated randomized proximal coordinate gradient (`APCG`) method to minimize composite convex functions.

As remarked earlier, [12], coupled with insights on NNLS+ problems provided in this work, can recover our guarantee for the optimality gap from Theorem 3.5. However, our work is the first to bring to the fore the properties of NNLS+ *required* to get such a guarantee, and our choice of primal-dual perspective allows for a stronger guarantee in terms of an upper bound on the primal-dual gap. Further, our algorithm is a novel type of acceleration, with our primal-dual perspective transparently illustrating our use of the aforementioned properties. We believe that these technical contributions, along with our techniques to obtain vastly improved theoretical guarantees with the restart strategy applied to this problem, are valuable to the broader optimization and machine learning communities.

## 2 Notation and preliminaries

Throughout the paper, we use bold lowercase letters to denote vectors and bold uppercase letters for matrices. For vectors and matrices, the operator $' \ge '$ is applied element-wise, and $\mathbb{R}_+$ is the non-negative part of the real line. We use $\langle \mathbf{a}, \mathbf{b} \rangle$ to denote the inner product of vectors $\mathbf{a}$ and $\mathbf{b}$ and $\nabla$

for gradient. Given a matrix $\mathbf{A}$, we use $\mathbf{A}_{:j}$ for its $j^{\text{th}}$ column vector, and for a vector $\mathbf{x}$, $x_j$ denotes its $j^{\text{th}}$ coordinate. We use $\text{nnz}(\mathbf{A})$ for the number of non-zero entries of $\mathbf{A}$. We use $\mathbf{x}_k$ for the vector in the $k^{\text{th}}$ iteration and, to disambiguate indexing, use $[\mathbf{x}_k]_j$ to mean the $j^{\text{th}}$ coordinate of $\mathbf{x}_k$. The $i^{\text{th}}$ standard basis vector is denoted by $\mathbf{e}_i$. For an $n$-dimensional vector $\mathbf{x}$ and $\mathbf{A} \in \mathbb{R}^{m \times n}$, we define $\mathbf{\Lambda} = \text{diag}([\|\mathbf{A}_{:1}\|_2^2, \ldots, \|\mathbf{A}_{:n}\|_2^2])$ and $\|\mathbf{x}\|_{\mathbf{\Lambda}}^2 = \sum_{i=1}^{n} x_i^2 \|\mathbf{A}_{:i}\|_2^2$. Finally, $[n] \overset{\text{def}}{=} \{1, 2, \ldots, n\}$.

A differentiable function $f : \mathbb{R}^n \to \mathbb{R}$ is convex if for any $\mathbf{x}, \widehat{\mathbf{x}} \in \mathbb{R}^n$, we have $f(\widehat{\mathbf{x}}) \geq f(\mathbf{x}) + \langle \nabla f(\mathbf{x}), \widehat{\mathbf{x}} - \mathbf{x} \rangle$. A differentiable function $f : \mathbb{R}^n \to \mathbb{R}$ is said to be $\mu$-strongly convex w.r.t. the $\ell_2$-norm if for any $\mathbf{x}, \widehat{\mathbf{x}} \in \mathbb{R}^n$, we have that $f(\widehat{\mathbf{x}}) \geq f(\mathbf{x}) + \langle \nabla f(\mathbf{x}), \widehat{\mathbf{x}} - \mathbf{x} \rangle + \frac{\mu}{2} \|\mathbf{x} - \widehat{\mathbf{x}}\|_2^2$. We have analogous definitions for concave and strongly concave functions, which flip the inequalities noted.

Given a convex program $\min_{\mathbf{x} \in \mathcal{X}} f(\mathbf{x})$, where $f : \mathbb{R}^n \to \mathbb{R}$ is differentiable and convex and $\mathcal{X} \subseteq \mathbb{R}^n$ closed and convex, the first-order optimality condition of a solution $\mathbf{x}^\star \in \text{argmin}_{\mathbf{x} \in \mathcal{X}} f(\mathbf{x})$ is

$$(\forall \mathbf{x} \in \mathcal{X}): \quad \langle \nabla f(\mathbf{x}^\star), \mathbf{x} - \mathbf{x}^\star \rangle \geq 0. \tag{1}$$

**Problem setup.** As discussed in the introduction, our goal is to solve (P), with $\mathbf{A} \in \mathbb{R}_+^{m \times n}$. For notational convenience, we work with the problem in the following scaled form:

$$\min_{\mathbf{x} \in \mathbb{R}_+^n} \left\{ f(\mathbf{x}) \overset{\text{def}}{=} \frac{1}{2} \|\mathbf{A}\mathbf{x}\|_2^2 - \mathbf{1}^\top \mathbf{x} \right\}, \tag{2}$$

This assumption is w.l.o.g. since any (P) can be brought to this form by a simple change of variable $\hat{x}_j = c_j x_j$ (see also e.g., [1, 14] for similar scaling ideas). The scaling need not be explicit in the algorithm since the change of variable $\hat{x}_j = c_j x_j$ is easily reversible.

**Properties of the objective.** To kick off our analysis, we highlight some properties inherent to the objective defined in (2). These properies, which strongly need the non-negativity of $\mathbf{A}$ and $\mathbf{x}$, are central to obtain a scale-invariant algorithm for (P).

**Proposition 2.1.** *Given* $f : \mathbb{R}_+^n \to \mathbb{R}$ *as defined in* (2) *and* $\mathbf{x}^\star \in \text{argmin}_{\mathbf{x} \in \mathbb{R}_+^n} f(\mathbf{x})$, *the following statements all hold.*

*a)* $\nabla f(\mathbf{x}^\star) \geq \mathbf{0}$.
*b)* $f(\mathbf{x}^\star) = -\frac{1}{2}\|\mathbf{A}\mathbf{x}^\star\|_2^2 = -\frac{1}{2}\mathbf{1}^\top \mathbf{x}^\star$.
*c) for all* $j \in [n]$, *we have* $x_j^\star \in \left[0, \frac{1}{\|\mathbf{A}_{:j}\|_2^2}\right]$.
*d)* $-\frac{1}{2} \sum_{j \in [n]} \frac{1}{\|\mathbf{A}_{:j}\|_2^2} \leq f(\mathbf{x}^\star) \leq -\frac{1}{2 \min_{j \in [n]} \|\mathbf{A}_{:j}\|_2^2}$.

The validity of division by $\|\mathbf{A}_{:j}\|_2$ in the preceding proposition is by the non-degeneracy of $\mathbf{A}$ discussed in the introduction. We prove this proposition in Section 6.1.

An important consequence of Proposition 2.1 (c) is that (P) can be restricted to the hyperrectangle $\mathcal{X} = \left\{\mathbf{x} \in \mathbb{R}^n : 0 \leq x_j \leq \frac{1}{\|\mathbf{A}_{:j}\|_2^2}\right\}$ without affecting its optimal solution, but effectively reducing the search space. Thus, going forward, we replace the constraint $\mathbf{x} \geq \mathbf{0}$ in (P) by $\mathbf{x} \in \mathcal{X}$.

**Primal-dual gap perspective.** As alluded earlier, our algorithm is analyzed through a primal-dual perspective. For this reason, it is useful to consider the Lagrangian

$$\mathcal{L}(\mathbf{x}, \mathbf{y}) = \langle \mathbf{A}\mathbf{x}, \mathbf{y} \rangle - \frac{1}{2}\|\mathbf{y}\|_2^2 - \mathbf{1}^\top \mathbf{x} \tag{3}$$

from which we can derive our rescaled problem (2) as the primal problem $\min_{\mathbf{x} \in \mathcal{X}} \mathcal{P}(\mathbf{x})$, where

$$\mathcal{P}(\mathbf{x}) = \max_{\mathbf{y} \in \mathbb{R}^m} \mathcal{L}(\mathbf{x}, \mathbf{y}) = -\mathbf{1}^\top \mathbf{x} + \max_{\mathbf{y} \geq \mathbf{0}} \left[ -\frac{1}{2}\|\mathbf{y}\|_2^2 + \langle \mathbf{A}\mathbf{x}, \mathbf{y} \rangle \right] = -\mathbf{1}^\top \mathbf{x} + \frac{1}{2}\|\mathbf{A}\mathbf{x}\|^2.$$

Thus, the Lagrangian is constructed in a way that one can derive the primal problem from it while also localizing the matrix in the bilinear term and ensuring coordinate-wise separability of the terms involving either only the dual or only the primal variable since this greatly simplifies our analysis. Similar to [45], we use Eq. (3) to define the following relaxation of the primal-dual gap, for arbitrary but fixed $\mathbf{u} \in \mathcal{X}$, and $\mathbf{v} \in \mathbb{R}^m$:

$$\text{Gap}_{\mathcal{L}}^{(\mathbf{u}, \mathbf{v})}(\mathbf{x}, \mathbf{y}) \overset{\text{def}}{=} \mathcal{L}(\mathbf{x}, \mathbf{v}) - \mathcal{L}(\mathbf{u}, \mathbf{y}). \tag{4}$$

The significance of this relaxed gap function is that for a candidate solution $\widetilde{\mathbf{x}}$ and an arbitrary $\widetilde{\mathbf{y}} \in \mathbb{R}^m$, a bound on $\mathrm{Gap}_{\mathcal{L}}^{(\mathbf{u},\mathbf{v})}(\widetilde{\mathbf{x}}, \widetilde{\mathbf{y}})$ translates to one on the primal error, as follows. First select $\mathbf{u} = \mathbf{x}^\star, \mathbf{v} = \mathbf{A}\widetilde{\mathbf{x}}$. Then, by observing that $\mathcal{L}(\widetilde{\mathbf{x}}, \mathbf{A}\widetilde{\mathbf{x}}) = f(\widetilde{\mathbf{x}})$ and $\mathcal{L}(\mathbf{x}^\star, \mathbf{A}\mathbf{x}^\star) = f(\mathbf{x}^\star)$, we have

$$f(\widetilde{\mathbf{x}}) - f(\mathbf{x}^\star) = \mathcal{L}(\widetilde{\mathbf{x}}, \mathbf{A}\widetilde{\mathbf{x}}) - \mathcal{L}(\mathbf{x}^\star, \mathbf{A}\mathbf{x}^\star).$$

For a fixed $\mathbf{x}$, $\mathcal{L}(\mathbf{x}, \cdot)$ is 1-strongly concave and minimized at $\mathbf{A}\mathbf{x}$. Thus, $\mathcal{L}(\mathbf{x}^\star, \mathbf{A}\widetilde{\mathbf{x}}) \leq \mathcal{L}(\mathbf{x}^\star, \mathbf{A}\mathbf{x}^\star) - \frac{1}{2}\|\mathbf{A}(\widetilde{\mathbf{x}} - \mathbf{x}^\star)\|^2$. Hence, we have the following primal bound:

$$f(\widetilde{\mathbf{x}}) - f(\mathbf{x}^\star) + \frac{1}{2}\|\mathbf{A}(\widetilde{\mathbf{x}} - \mathbf{x}^\star)\|^2 \leq \mathcal{L}(\widetilde{\mathbf{x}}, \mathbf{A}\widetilde{\mathbf{x}}) - \mathcal{L}(\mathbf{x}^\star, \widetilde{\mathbf{y}}) = \mathrm{Gap}_{\mathcal{L}}^{(\mathbf{x}^\star, \mathbf{A}\widetilde{\mathbf{x}})}(\widetilde{\mathbf{x}}, \widetilde{\mathbf{y}}). \tag{5}$$

In light of this connection, our algorithm for bounding the primal error is one that generates iterates that can be used to construct bounds on $\mathrm{Gap}_{\mathcal{L}}^{(\mathbf{u},\mathbf{v})}(\widetilde{\mathbf{x}}, \widetilde{\mathbf{y}})$, as we detail next.

## 3 Our algorithm and convergence analysis

Our algorithm, Scale Invariant NNLS+ (SI-NNLS+), is an iterative algorithm using the estimate sequences $\phi_k(\mathbf{x})$ and $\psi_k(\mathbf{y})$ for $k \geq 1$ (see Section 3.1) giving the primal and dual updates

$$\mathbf{x}_k = \underset{\mathbf{x} \in \mathcal{X}}{\operatorname{argmin}} \, \phi_k(\mathbf{x}) \quad \text{and} \quad \mathbf{y}_k = \underset{\mathbf{y} \in \mathbb{R}^m}{\operatorname{argmax}} \, \psi_k(\mathbf{y}). \tag{6}$$

We use our algorithm's iterates from Eq. (6) to construct $\mathrm{G}_k$, an upper estimate of $\mathrm{Gap}_{\mathcal{L}}^{(\mathbf{u},\mathbf{v})}(\widetilde{\mathbf{x}}_k, \widetilde{\mathbf{y}}_k)$, where $\widetilde{\mathbf{x}}_k, \widetilde{\mathbf{y}}_k$ are convex combinations of the iterates and $\mathbf{u} \in \mathcal{X}, \mathbf{v} \in \mathbb{R}_+^m$, where $\mathbf{u}$ is fixed and $\mathbf{v}$ is arbitrary. Our motivation for constructing $\mathrm{G}_k(\mathbf{u}, \mathbf{v}) \geq \mathrm{Gap}_{\mathcal{L}}^{(\mathbf{u},\mathbf{v})}(\widetilde{\mathbf{x}}_k, \widetilde{\mathbf{y}}_k)$ is to obtain a bound on the primal error, using Inequality (5). The main goal in the analysis is to show that $\mathrm{G}_k \leq \frac{Q}{A_k}$, where $A_k$ is a sequence of positive numbers and $Q$ is bounded. To obtain the claimed multiplicative approximation, we use Inequality (5) and argue that for $\mathbf{u} = \mathbf{x}^\star, \mathbf{v} = \mathbf{A}\widetilde{\mathbf{x}}_k$, we have $Q \leq O(|f(\mathbf{x}^*)|)$.

Our algorithm may be interpreted as a variant of the VRPDA$^2$ algorithm [45], with our analysis inspired by the approximate duality gap technique [13]; however, unlike VRPDA$^2$, which uses estimate sequences, our analysis directly bounds the primal-dual gap. Another difference from VRPDA$^2$ is in our choice of primal regularizer (described shortly) and our lack of a dual regularizer. A variant of SI-NNLS+ suitable for analysis is shown in Algorithm 1, with proofs in Section 6.2 and Section 6.3. We give an equivalent implementation version ("*Lazy* SI-NNLS+", which updates as few coordinates as is possible to improve cost per iteration) in Algorithm 2 and its analysis in Section 7.

---

**Algorithm 1** Scale Invariant Non-negative Least Squares with Non-negative Data (SI-NNLS+)

---

1: **Input:** Matrix $\mathbf{A} \in \mathbb{R}_+^{m \times n}$ with $n \geq 4$, accuracy $\varepsilon$, initial point $\mathbf{x}_0$
2: Initialize: $\widetilde{\mathbf{x}}_0 = \mathbf{x}_0, \overline{\mathbf{y}}_0 = \mathbf{y}_0 = \mathbf{A}\mathbf{x}_0$, $K = \frac{5}{2}n\log n + \frac{6n}{\sqrt{\varepsilon}}$, $a_1 = \frac{1}{\sqrt{2}n^{1.5}}$, $a_2 = \frac{a_1}{n-1}$, $A_0 = 0$,
   $A_1 = a_1$, $\phi_0(\mathbf{x}) = \frac{1}{2}\|\mathbf{x} - \mathbf{x}_0\|_{\mathbf{\Lambda}}^2$.
3: **for** $k = 1, 2, \ldots, K$ **do**
4:     Sample $j_k$ uniformly at random from $\{1, 2, \ldots, n\}$
5:     $\mathbf{x}_k \leftarrow \arg\min_{\mathbf{x} \in \mathcal{X}} \phi_k(\mathbf{x})$, for $\phi_k(\mathbf{x})$ defined by Eq. (12) and Eq. (15)
6:     $\mathbf{y}_k \leftarrow \arg\max_{\mathbf{y} \in \mathbb{R}^m} \psi_k(\mathbf{y})$, for $\psi_k(\mathbf{y})$ defined by Eq. (9)
7:     $\widetilde{\mathbf{x}}_k = \frac{1}{A_k}\left[A_{k-1}\widetilde{\mathbf{x}}_{k-1} + a_k\left(n\mathbf{x}_k - (n-1)\mathbf{x}_{k-1}\right)\right]$.
8:     $\overline{\mathbf{y}}_k \leftarrow \mathbf{y}_k + \frac{a_k}{a_{k+1}}(\mathbf{y}_k - \mathbf{y}_{k-1})$
9:     $A_{k+1} \leftarrow A_k + a_{k+1}$, $a_{k+2} = \min\{\frac{na_{k+1}}{n-1}, \frac{\sqrt{A_{k+1}}}{2n}\}$
10: **end for**
11: Return $\widetilde{\mathbf{x}}_K$

---

### 3.1 Gap estimate construction

The gap estimate $\mathrm{G}_k$ is constructed as the difference $\mathrm{G}_k(\mathbf{u}, \mathbf{v}) = \mathrm{U}_k(\mathbf{v}) - \mathrm{L}_k(\mathbf{u})$, where $\mathrm{U}_k(\mathbf{v}) \geq \mathcal{L}(\widetilde{\mathbf{x}}_k, \mathbf{v})$ and $\mathrm{L}_k(\mathbf{u}) \leq \mathcal{L}(\mathbf{u}, \widetilde{\mathbf{y}}_k)$ are, respectively, upper and lower bounds we construct on the Lagrangian. It then follows by Eq. (4) that $\mathrm{G}_k(\mathbf{u}, \mathbf{v})$ is a valid upper estimate of $\mathrm{Gap}_{\mathcal{L}}^{(\mathbf{u},\mathbf{v})}(\widetilde{\mathbf{x}}_k, \widetilde{\mathbf{y}}_k)$.

We first introduce a technical component our constructions $\mathrm{L}_k$ and $\mathrm{U}_k$ crucially hinge on: we define two positive sequences of numbers $\{a_i\}_{i \geq 1}$ and $\{a_i^k\}_{1 \leq i \leq k}$, with one of their properties being that both sum up to $A_k > 0$ for $k \geq 1$. Specifically, we define $A_0 = 0$ and $\{a_i\}_{i \geq 1}$ as $a_i = A_i - A_{i-1}$. The sequence $\{a_i^k\}$ changes with $k$ and for $k = 1$ is defined by $a_1^1 = a_1$, while for $k \geq 2$ :

$$a_i^k = \begin{cases} a_1 - (n-1)a_2, & \text{if } i = 1, \\ na_i - (n-1)a_{i+1}, & \text{if } 2 \leq i \leq k-1, \\ na_k, & \text{if } i = k. \end{cases} \tag{7}$$

Summing over $i \in [k]$ verifies that $A_k = \sum_{i=1}^{k} a_i^k$. For the sequence $\{a_i^k\}_{1 \leq i \leq k}$ to be non-negative, we further require that $a_1 - (n-1)a_2 \geq 0$ and $\forall i \geq 2, na_i - (n-1)a_{i+1} \geq 0$.

The significance of these two sequences lies in defining the algorithm's primal-dual output pair by

$$\widetilde{\mathbf{x}}_k = \frac{1}{A_k} \sum_{i \in [k]} a_i^k \mathbf{x}_i \quad \text{and} \quad \widetilde{\mathbf{y}}_k = \frac{1}{A_k} \sum_{i \in [k]} a_i \mathbf{y}_i. \tag{8}$$

The intricate interdependence of $\{a_i\}$ and $\{a_i^k\}$ enables expressing $\widetilde{\mathbf{x}}_k$ in terms of only $\{a_i\}$. This expression further simplifies to a cheaper recursive one, which is used in Algorithm 1.

With the sequences $\{a_i\}_{i \geq 1}$ and $\{a_i^k\}_{1 \leq i \leq k}$ in tow, we are now ready to show the construction of an upper bound $\mathrm{U}_k(\mathbf{v})$ on $\mathcal{L}(\widetilde{\mathbf{x}}_k, \mathbf{v})$ and a lower bound $\mathrm{L}_k(\mathbf{u})$ on $\mathcal{L}(\mathbf{u}, \widetilde{\mathbf{y}}_k)$.

**Upper bound.** To construct an upper bound, first observe that by Eq. (3) and Eq. (8),

$$\mathcal{L}(\widetilde{\mathbf{x}}_k, \mathbf{v}) = \langle \mathbf{A}\widetilde{\mathbf{x}}_k, \mathbf{v} \rangle - \frac{1}{2}\|\mathbf{v}\|_2^2 - \mathbf{1}^\top \widetilde{\mathbf{x}}_k = \frac{1}{A_k} \sum_{i \in [k]} a_i^k \left[ \langle \mathbf{A}\mathbf{x}_i, \mathbf{v} \rangle - \frac{1}{2}\|\mathbf{v}\|_2^2 - \mathbf{1}^\top \mathbf{x}_i \right].$$

Consider the primal estimate sequence defined for $k = 0$ as $\psi_0 = 0$ and for $k \geq 1$ by

$$\psi_k(\mathbf{v}) \stackrel{\text{def}}{=} \sum_{i \in [k]} a_i^k \left[ \langle \mathbf{A}\mathbf{x}_i, \mathbf{v} \rangle - \frac{1}{2}\|\mathbf{v}\|_2^2 - \mathbf{1}^\top \mathbf{x}_i \right], \tag{9}$$

which ensures that $\mathcal{L}(\widetilde{\mathbf{x}}_k, \mathbf{v}) = \frac{1}{A_k}\psi_k(\mathbf{v})$. A key upshot of constructing $\psi_k(\mathbf{v})$ as in Eq. (9) is that the quadratic term implies $A_k$-strong concavity of $\psi_k$ for $k \geq 1$, which in turn ensures that the vector $\mathbf{y}_k = \arg\max_{\mathbf{y} \in \mathbb{R}^m} \psi_k(\mathbf{y})$ from Eq. (6) is unique. This property, coupled with the first-order optimality condition in Inequality (1), gives that for any $\mathbf{y} \in \mathbb{R}^m$, $\psi_k(\mathbf{y}) \leq \psi_k(\mathbf{y}_k) - \frac{A_k}{2}\|\mathbf{y} - \mathbf{y}_k\|_2^2$. We are now ready to define the following upper bound by:

$$\mathrm{U}_k(\mathbf{v}) \stackrel{\text{def}}{=} \frac{1}{A_k}\psi_k(\mathbf{y}_k) - \frac{1}{2}\|\mathbf{v} - \mathbf{y}_k\|_2^2. \tag{10}$$

The preceding discussion immediately implies that $\mathrm{U}_k$ is a valid upper bound for the Lagrangian.

**Lemma 3.1.** *For $\mathrm{U}_k$ as defined in Eq. (10), Lagrangian defined in Eq. (3) and $\widetilde{\mathbf{x}}_k \in \mathbb{R}_+^n$ in Eq. (8), we have, for all $\mathbf{y} \in \mathbb{R}^m$, the upper bound $\mathrm{U}_k(\mathbf{y}) \geq \mathcal{L}(\widetilde{\mathbf{x}}_k, \mathbf{y})$.*

**Lower bound.** Analogous to the preceding section, we now obtain a *lower bound* on the Lagrangian, completing the bound on the gap estimate. However, the construction becomes more technical. We start with the same approach as for the upper bound. Since $\mathcal{L}(\mathbf{u}, \widetilde{\mathbf{y}}_k)$ is concave in $\widetilde{\mathbf{y}}_k$, by Jensen's inequality: $\mathcal{L}(\mathbf{u}, \widetilde{\mathbf{y}}_k) \geq \frac{1}{A_k} \sum_{i \in [k]} a_i \left( \langle \mathbf{A}\mathbf{u}, \mathbf{y}_i \rangle - \mathbf{1}^\top \mathbf{u} - \frac{1}{2}\|\mathbf{y}_i\|_2^2 \right)$. Were we to define the dual estimate sequence $\phi_k$ in the same way as we did for the primal estimate sequence $\psi_k$, we would now simply define it as $A_k$ times the right-hand side in the last inequality. However, doing so would make $\phi_k$ depend on $\mathbf{y}_k$, which is updated *after* $\mathbf{x}_k$, which in Eq. (6) is defined as the minimizer of the $\phi_k$.

To avoid such a circular dependency, we add and subtract a linear term $\sum_{i \in [k]} \langle \mathbf{A}^\top \overline{\mathbf{y}}_{i-1}, \mathbf{u} \rangle$, where $\overline{\mathbf{y}}_{i-1}$, defined later, are extrapolation points that depend only on $\mathbf{y}_1, \dots \mathbf{y}_{i-1}$. We thus have

$$\mathcal{L}(\mathbf{u}, \widetilde{\mathbf{y}}_k) \geq \frac{1}{A_k} \sum_{i \in [k]} a_i \left[ \langle \mathbf{A}\mathbf{u}, \overline{\mathbf{y}}_{i-1} \rangle - \mathbf{1}^\top \mathbf{u} - \frac{1}{2}\|\mathbf{y}_i\|_2^2 \right] + \frac{1}{A_k} \sum_{i \in [k]} a_i \langle \mathbf{A}\mathbf{u}, \mathbf{y}_i - \overline{\mathbf{y}}_{i-1} \rangle.$$

If we now defined $\phi_k$ based on the first term in the above inequality, we run into another obstacle: the linearity of the resulting estimate sequence is insufficient for cancelling all the error terms in the analysis. Hence, as is common, we introduce strong convexity by adding and subtracting an appropriate strongly convex function. Our chosen strongly convex function is motivated by the box-constrained property of the optimum from Proposition 2.1 (c) and crucial in bounding the initial gap estimate. It coincides with $\phi_0$: for any $\mathbf{x} \in \mathbb{R}_+^n$, define the function

$$\phi_0(\mathbf{x}) = \frac{1}{2}\|\mathbf{x} - \mathbf{x}_0\|_{\boldsymbol{\Lambda}}^2. \tag{11}$$

This function is 1-strongly convex with respect to $\|\cdot\|_{\boldsymbol{\Lambda}}$ and used in defining $\phi_1$ as:

$$\phi_1(\mathbf{u}) = a_1\langle \mathbf{A}^\top \overline{\mathbf{y}}_0 - \mathbf{1}, \mathbf{u}\rangle + \phi_0(\mathbf{u}). \tag{12}$$

The definition of $\phi_1$ is driven by the purpose of cancelling initial error terms. Next, we choose $\phi_k$ so that for any fixed $\mathbf{u} \in \mathcal{X}$, we have

$$\mathbb{E}[\phi_k(\mathbf{u})] = \mathbb{E}\Big[\sum_{i \in [k]} a_i\langle \mathbf{A}^\top \overline{\mathbf{y}}_{i-1} - \mathbf{1}, \mathbf{u}\rangle + \phi_0(\mathbf{u})\Big], \tag{13}$$

where the expectation is with respect to all the randomness in the algorithm. This construction is used to reduce the per-iteration complexity, for which we employ a *randomized* coordinate update on $\mathbf{x}_k$ for $k \geq 2$. To support such updates, we relax the lower bound to hold only *in expectation*.

Concretely, let $j_i$ be the coordinate sampled uniformly at random from $[n]$ in the $i^{\text{th}}$ iteration of SI-NNLS+, independent of history. Fix $\mathbf{y}_i$ for $i = 1, \ldots, k-1$ and for $k \geq 2$ and $\mathbf{x} \in \mathcal{X}$, define

$$\phi_k(\mathbf{x}) = \phi_1(\mathbf{x}) + \sum_{i=2}^k na_i\langle \mathbf{A}^\top \overline{\mathbf{y}}_{i-1} - \mathbf{1}, x_{j_i}\mathbf{e}_{j_i}\rangle. \tag{14}$$

For $k \geq 2$, $\phi_k(\mathbf{u})$ can also be defined recursively via

$$\phi_k(\mathbf{x}) = \phi_{k-1}(\mathbf{x}) + na_k\langle \mathbf{A}^\top \overline{\mathbf{y}}_{k-1} - \mathbf{1}, x_{j_k}\mathbf{e}_{j_k}\rangle. \tag{15}$$

The function $\phi_k$ inherits the strong convexity of $\phi_0$. This property, together with Eq. (6) and first-order optimality from Inequality (1), give

$$\phi_k(\mathbf{x}) \geq \phi_k(\mathbf{x}_k) + \frac{1}{2}\|\mathbf{x} - \mathbf{x}_k\|_{\boldsymbol{\Lambda}}^2. \tag{16}$$

Along with strong convexity, our choice of $\phi_k$ in Eq. (15) leads to the following properties essential to our analysis: (1) $\phi_k$ is separable in its coordinates; (2) the primal variable $\mathbf{x}_k$ is updated only at its $j_k^{\text{th}}$ coordinate; (3) Eq. (13) is true. These are formally stated in Proposition 6.2.

With the dual estimate sequence $\phi_k$ defined in Eq. (15), we now define the sequence $\mathrm{L}_k$ by

$$\mathrm{L}_k(\mathbf{x}) \stackrel{\text{def}}{=} \frac{\phi_k(\mathbf{x}_k) + \frac{1}{2}\|\mathbf{x} - \mathbf{x}_k\|_{\boldsymbol{\Lambda}}^2 - \phi_0(\mathbf{x}) + \sum_{i \in [k]} a_i(\langle \mathbf{A}\mathbf{x}, \mathbf{y}_i - \overline{\mathbf{y}}_{i-1}\rangle - \frac{1}{2}\|\mathbf{y}_i\|_2^2)}{A_k}. \tag{17}$$

We conclude this section by justifying our choice of $\mathrm{L}_k$ as a valid expected lower bound on $\mathbb{E}\mathcal{L}(\mathbf{x}, \widetilde{\mathbf{y}}_k)$.

**Lemma 3.2.** *For $\mathrm{L}_k$ defined in Eq. (17), for the Lagrangian in Eq. (3) and $\widetilde{\mathbf{y}}_k$ in Eq. (8), we have, for a fixed $\mathbf{u} \in \mathcal{X}$, the lower bound $\mathbb{E}\mathcal{L}(\mathbf{u}, \widetilde{\mathbf{y}}_k) \geq \mathbb{E}\mathrm{L}_k(\mathbf{u})$, where the expectation is with respect to all the random choices of coordinates in Algorithm 1.*

### 3.2 Bounding the gap estimate

With the gap estimate $\mathrm{G}_k$ constructed as in the preceding section and combining Eq. (10) and Eq. (17), we now achieve our goal of bounding $A_k\mathrm{G}_k$ (to obtain a convergence rate of the order $1/A_k$) by bounding the change in $A_k\mathrm{G}_k$ and the initial scaled gap $A_1\mathrm{G}_1$.

**Lemma 3.3.** *Consider the iterates $\{\mathbf{x}_k\}$ and $\{\mathbf{y}_k\}$ evolving according to Algorithm 1. Let $n \geq 2$ and assume that $a_1 = \frac{1}{\sqrt{2}n^{1.5}}$ and $a_1 \geq (n-1)a_2$, while for $k \geq 3$,*

$$a_k \leq \min\Big(\frac{na_{k-1}}{n-1}, \frac{\sqrt{A_{k-1}}}{2n}\Big). \tag{18}$$

*Then, for fixed $\mathbf{u} \in \mathcal{X}$, any $\mathbf{v} \in \mathbb{R}^m$, and all $k \geq 2$, the gap estimate $\mathrm{G}_k = \mathrm{U}_k - \mathrm{L}_k$ satisfies*

$$\mathbb{E}(A_k G_k(\mathbf{x}, \mathbf{y}) - A_{k-1} G_{k-1}(\mathbf{x}, \mathbf{y}))$$

$$\leq -\mathbb{E}\left(\frac{A_k}{2}\|\mathbf{y} - \mathbf{y}_k\|_2^2 - \frac{A_{k-1}}{2}\|\mathbf{y} - \mathbf{y}_{k-1}\|_2^2\right) - \frac{1}{2}\mathbb{E}\|\mathbf{x} - \mathbf{x}_k\|_\Lambda^2 + \frac{1}{2}\mathbb{E}\|\mathbf{x} - \mathbf{x}_{k-1}\|_\Lambda^2$$

$$- a_k \mathbb{E}\langle \mathbf{A}(\mathbf{x} - \mathbf{x}_k), \mathbf{y}_k - \mathbf{y}_{k-1}\rangle + a_{k-1}\mathbb{E}\langle \mathbf{A}(\mathbf{x} - \mathbf{x}_{k-1}), \mathbf{y}_{k-1} - \mathbf{y}_{k-2}\rangle$$

$$- \frac{1}{4}A_{k-1}\mathbb{E}\|\mathbf{y}_k - \mathbf{y}_{k-1}\|_2^2 + \frac{1}{4}A_{k-2}\mathbb{E}\|\mathbf{y}_{k-1} - \mathbf{y}_{k-2}\|_2^2.$$

**Lemma 3.4.** *Given a fixed $\mathbf{u} \in \mathcal{X}$, any $\mathbf{v} \in \mathbb{R}^m$, $\bar{\mathbf{y}}_0 = \mathbf{y}_0$, and $\mathbf{x}_1$ and $\mathbf{y}_1$ from Algorithm 1, we have*

$$A_1 \mathrm{G}_1(\mathbf{u}, \mathbf{v}) = a_1 \langle \mathbf{A}^\top(\mathbf{y}_1 - \mathbf{y}_0), \mathbf{x}_1 - \mathbf{u}\rangle + \phi_0(\mathbf{u}) - \phi_0(\mathbf{x}_1) - \frac{1}{2}\|\mathbf{u} - \mathbf{x}_1\|_\Lambda^2 - \frac{A_1}{2}\|\mathbf{v} - \mathbf{y}_1\|_2^2.$$

Combining the two lemmas, we now bound $G_K$ and deduce our final result on the primal error.

**Theorem 3.5.** *[Main Result] Assume that $n \geq 4$. Given a matrix $\mathbf{A} \in \mathbb{R}_+^{m \times n}$, $\varepsilon > 0$, an arbitrary $\mathbf{x}_0 \in \mathcal{X}$ and $\bar{\mathbf{y}}_0 = \mathbf{y}_0 = \mathbf{A}\mathbf{x}_0$, let $\mathbf{x}_k$ and $A_k$ evolve according to SI-NNLS+ (Algorithm 1) for $k \geq 1$. For $f$ defined in (2), define $\mathbf{x}^\star \in \arg\min_{\mathbf{x} \geq \mathbf{0}} f(\mathbf{x})$. Then, for all $K \geq 2$, we have*

$$\mathbb{E}\left[\langle \nabla f(\widetilde{\mathbf{x}}_K), \widetilde{\mathbf{x}}_K - \mathbf{x}^\star\rangle + \frac{1}{2}\|\mathbf{A}(\widetilde{\mathbf{x}}_K - \mathbf{x}^\star)\|^2\right] \leq \frac{2\phi_0(\mathbf{x}^\star)}{A_K} = \frac{\|\mathbf{x}_0 - \mathbf{x}^\star\|_\Lambda^2}{A_K}.$$

*When $K \geq \frac{5}{2}n \log n$, we have $A_K \geq \frac{(K - \frac{5}{2}n \log n)^2}{36n^2}$. If $\phi_0(\mathbf{x}^\star) \leq |f(\mathbf{x}^\star)|$, then for $K \geq \frac{5}{2}n \log n + \frac{6n}{\sqrt{\varepsilon}}$, we have $\mathbb{E}[f(\widetilde{\mathbf{x}}_K) - f(\mathbf{x}^\star)] \leq \varepsilon|f(\mathbf{x}^\star)|$. The total cost is $O\big(nnz(\mathbf{A})\big(\log n + \frac{1}{\sqrt{\varepsilon}}\big)\big)$.*

The assumption $\phi_0(\mathbf{x}^\star) \leq |f(\mathbf{x}^\star)|$ above is satisfied by $\mathbf{x}_0 = \mathbf{0}$ (c.f. Proposition 2.1 and Eq. (11)). We reiterate that the reason $\|\mathbf{A}\|$ does not show up in the final bounds (thereby rendering our algorithm "scale-invariant") is because Proposition 2.1 allows bounding $\|\mathbf{x}_0 - \mathbf{x}^\star\|_\Lambda^2$ by $|f(\mathbf{x}^\star)|$, where we crucially used the non-negativity of $\mathbf{A}$ and $\mathbf{x}$; this does not seem possible for general $\mathbf{A}$.

*Remark* 3.6. SI-NNLS+ (Algorithm 1) and Theorem 3.5 also generalize to a mini-batch version. Increasing the batch size grows our bounds and number of data passes by a factor of at most square-root of the batch size $s$, by relating the spectral norm of the $s$ columns of $\mathbf{A}$ corresponding to a batch to the Euclidean norms of individual columns of $\mathbf{A}$ from the same batch. However, due to efficient available implementations of vector operations, mini-batch variants of our algorithm with small batch sizes can have lower total runtimes on some datasets (see Section 5).

## 4 Adaptive restart

We now describe how SI-NNLS+ can be combined with adaptive restart to obtain linear convergence rate. To apply the restart strategy, we need suitable upper and lower bounds on the measure of convergence rate. Our measure of optimality is the *natural residual* $r(\mathbf{x}) = \|\mathbf{R}(\mathbf{x})\|_\Lambda$ [31] for

$$\mathbf{R}(\mathbf{x}) = \mathbf{x} - \Pi_{\mathbb{R}_+^n}(\mathbf{x} - \Lambda^{-1}\nabla f(\mathbf{x})) = \mathbf{x} - (\mathbf{x} - \Lambda^{-1}\nabla f(\mathbf{x}))_+, \tag{19}$$

where $\Pi_{\mathbb{R}_+^n}$ is the projection operator onto $\mathbb{R}_+^n$ and $\Lambda$ is as defined in Section 2. For $\Lambda = \mathbf{I}$, $\mathbf{R}(\mathbf{x})$ is the natural map as defined in, e.g., [15]. Due to space constraints, we only state the main result of this section in the following theorem, while full technical details are deferred to Section 6.4.

**Theorem 4.1.** *Given an error parameter $\varepsilon > 0$ and $\mathbf{x}_0 = \mathbf{0}$, consider the following algorithm $\mathcal{A}$ :*

---

$\mathcal{A}$ : ***SI-NNLS+ with Restarts***
*Initialize: $k = 1$.*
*Initialize Lazy SI-NNLS+ at $\mathbf{x}_{k-1}$.*
*Run Lazy SI-NNLS+ until the output $\widetilde{\mathbf{x}}_K^k$ satisfies $r(\widetilde{\mathbf{x}}_K^k) \leq \frac{1}{2}r(\mathbf{x}_{k-1})$.*
*Restart Lazy SI-NNLS+ initializing at $\mathbf{x}_k = \widetilde{\mathbf{x}}_K^k$.*
*Increment $k$.*
*Repeat until $r(\widetilde{\mathbf{x}}_K^k) \leq \varepsilon$.*

---

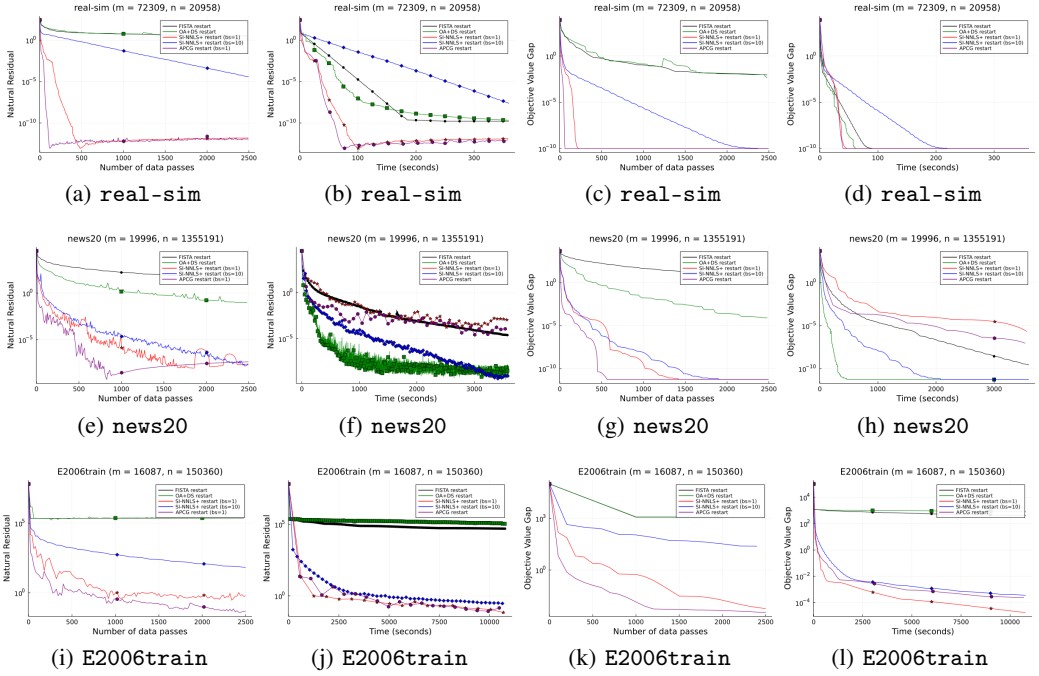

Figure 1: NNLS+ algorithms with restart on `real-sim`, `news20` and `E2006train` with spectral norm $3 \cdot 10^{-3}, 10^{-3}, 5 \cdot 10^{-6}$ and condition number $6 \cdot 10^{18}, 5 \cdot 10^{10}, 6 \cdot 10^{5}$, respectively.

*Then, the expected number of arithmetic operations of $\mathcal{A}$ is $O\big(\mathrm{nnz}(\mathbf{A})\big(\log n + \frac{\sqrt{n}}{\mu}\big)\log\big(\frac{r(\mathbf{x}_0)}{\varepsilon}\big)\big)$. As a consequence, given $\bar{\varepsilon} > 0$, the total expected number of arithmetic operations until a point with $f(\mathbf{x}) - f(\mathbf{x}^\star) \le \bar{\varepsilon}|f(\mathbf{x}^\star)|$ can be constructed by $\mathcal{A}$ is $O\Big(\mathrm{nnz}(\mathbf{A})\Big(\log n + \frac{\sqrt{n}}{\mu}\Big)\log\Big(\frac{n}{\mu\bar{\varepsilon}}\Big)\Big)$.*

## 5 Numerical experiments and discussion

We conclude our paper by presenting the numerical performance of SI-NNLS+ and its restart versions (see the efficient implementation version in Algorithm 2) against FISTA with restart [5, 38], a general-purpose large-scale optimization algorithm, OA+DS with restart designed by [23] specifically for large-scale NNLS problems, and lazy implemented APCG [16] with restart. We use the same restart strategy for all the algorithms, proposed in Section 4. As an accelerated algorithm, FISTA has the optimal $1/k^2$ convergence rate; OA+DS, while often efficient in practice, has only an asymptotic convergence guarantee. For FISTA, we compute the tightest Lipschitz constant (i.e., the spectral norm $\|\mathbf{A}\|$); for OA+DS, we follow the best practices laid out by [23]. For our SI-NNLS+ algorithm and its restart version with batch size bs $= 1$, we follow Algorithm 2 and the restart strategy in Section 4.[4] For the restart version with batch size larger than 1, we choose the best batch size in $\{10, 50, 300, 500\}$ and compute the block coordinate Lipschitz constants as the spectral norms of the corresponding block matrices. All algorithms were implemented in Julia and run on a server with 32 Intel(R) Xeon(R) Silver 4110 32-Core Processors.

We evaluated the performance of the algorithms on the large-scale sparse datasets `real-sim`, `news20`, and `E2006train` from the LibSVM library [9]. Both `real-sim` and `news20` datasets have non-negative data matrices, but the labels may be negative. When there exist negative labels, it is possible for the elements of $\mathbf{A}^\top \mathbf{b}$ to be negative. In such cases, per the discussion from the introduction, we can simply remove the corresponding columns of $\mathbf{A}$ and solve an equivalent problem with smaller dimension. On the other hand, the data matrix in `E2006train` dataset is not non-negative, which means that this dataset does not satisfy the assumption required for the analysis of Algorithm 1.

---

[4]The algorithm is implemented for the non-scaled version of the problem, (P); see Section 2. An implementation is in https://github.com/arcturus611/nnlr-2021.

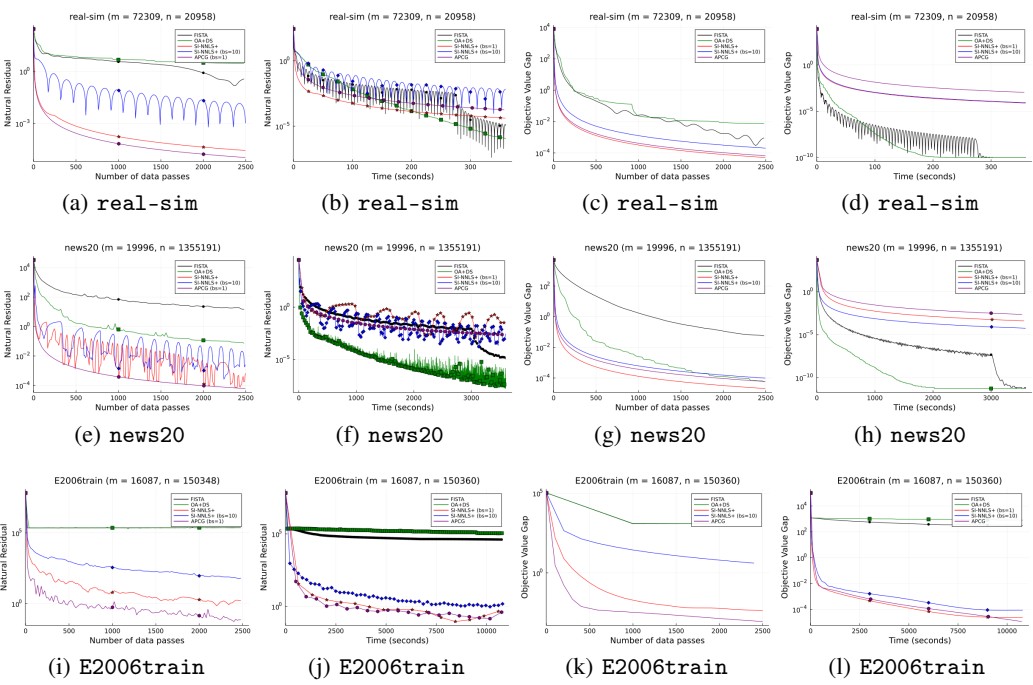

Figure 2: NNLS+ algorithms without restart on `real-sim`, `news20` and `E2006train` with spectral norm $3 \cdot 10^{-3}, 10^{-3}, 5 \cdot 10^{-6}$ and condition number $6 \cdot 10^{18}, 5 \cdot 10^{10}, 6 \cdot 10^5$, respectively.

However, Algorithm 1 can still be run by keeping only the non-negativity constraints for primal updates. This example is provided solely for illustration of empirical performance.

**Results.** To compare all implemented algorithms, we plot the natural residual/objective value gap versus number of data passes/time in Figure 1 for all algorithms implemented with restart and in Figure 2 for all algorithms implemented without restart. As can be observed from the two figures, our proposed restart speeds up all the algorithms. Figure 1(a)-(d) shows that SI-NNLS+ is better than FISTA and OA+DS in terms of number of data passes on the `real-sim` dataset and better than APCG in terms of time. While the proposed restart strategy speeds up all the algorithms to linear convergence, variants of SI-NNLS+ remain competitive in all the settings. In terms of the performance of different variants of SI-NNLS+, with bs = 1, we have a much better coordinate Lipschitz constant than for bs = 10 and thus the case of bs = 1 dominates bs = 10 in terms of data passes. As FISTA and OA+DS take less time accessing the full dataset once, they have lower runtimes than SI-NNLS+ but are beaten by SI-NNLS+ with restart and bs = 1. In Figure 1(e)-(h), on the `news20` dataset, in terms of number of data passes, restarted APCG and SI-NNLS+ with bs = 1 are dominant. However, as `news20` is a very sparse dataset, letting bs = 1 significantly increases the total time to access the full data once due to the overhead per iteration. As a result, single-coordinate methods have the worst runtimes, while restarted OA+DS is the fastest but SI-NNLS+ with bs = 10 remains competitive. Figure 1(i)-(l) shows the performance comparison on the `E2006train` dataset. On this dataset, both restarted FISTA and restarted OA+DS ran for 4 hours without visibly reducing the function value. SI-NNLS+ outperforms FISTA and OA+DS in both number of data passes and time, and outperforms APCG in terms of time. Further experiments are left for future work.

## Acknowledgements

CS was supported in part by the NSF grant 2023239. JD was supported in part by the NSF grant 2007757, by the Office of Naval Research under contract number N00014-22-1-2348, and by the Wisconsin Alumni Research Foundation. Part of this work was done while JD was visiting Simons Institute for the Theory of Computing. Part of this work was done when SP was visiting JD at the University of Wisconsin-Madison.

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
