# OpenReview forum: "A Fast Scale-Invariant Algorithm for Non-negative Least Squares with Non-negative Data"
_NeurIPS.cc/2022/Conference — NeurIPS 2022 Accept_

### Official Review · Reviewer_5CSB · 2022-07-03

**Rating:** 7
**Confidence:** 3
**Soundness:** 3 good
**Presentation:** 3 good
**Contribution:** 3 good

**Summary:**

The paper considers the classical non-negative least squares problem, and makes an observation that when the
data itself is also non-negative, this considerably simplifies the problem.  It proposes an algorithm inspired
by primal-dual variance reduction, and establishes scale-invariant (not dependent on matrix norms) multiplicative
convergence guarantees, and linear convergence using an adaptive restart technique.

**Questions:**

7) You report convergence in terms of natural residual.  Why is this a good measure (beyond being convenient for analysis).  It's worth
commenting how well it translates to other standard convergence metric -- especially for the experimental section. Does your method work especially well under the lens of natural residual?

8) Algorithm 2 is not defined -- is that the algorithm in Theorem 4.1?

9) For restart versions we choose the best batch size in {10,50,300,500}.  For a solver to be used in practice -- how do you know (without
running all versions) which batch size to use -- any heuristics?

**Limitations:**

I would've liked to see more discussion of limitations and more thorough comparison with competing approaches.

**Strengths And Weaknesses:**

Key strengths of the paper:
* pointing out that NNLS is significantly easier with non-negative data (called NNLS+ in the paper).
* Developing a new optimization algorithms for NNLS+, and its detailed convergence analysis.
* Numerical experiments show cases with significant improvement upon existing  methods (some comments below).
* The paper is clearly written.

Weaknesses:

1) I'm not fully convinced of the prevalence of NNLS+ (as opposed to plain NNLS) in applications. Taking NNLS+ and
simply subtracting the mean, de-trending, or removing first principal component (or any kind of residual) will in general induce
negative values in the data. Most practical applications of NNLS I've seen included some negative data. Even in the
experiments both examples included negative data (1st example is indeed trivially addressed, while 2nd is not NNLS+).
Also, since you applied the approach to plain NNLS in experiments -- can you comment on whether you expect the approach
to be competitive for NNLS at large?

2) As you mentioned,  an important application of NNLS+ is to NMF.  Is the proposed approach competitive w.r.t. state-of-the-art
NMF solvers? (Are these based on alternating NNLS -- or are there other approaches as well?)

3) It's worth mentioning other classical approaches to deal with non-negaitve targets, e.g. generalized linear models (e.g. Poisson
regression).

4) The paper claims that plain LS necessarily scales with matrix constants (e.g. spectral norms). In addition to iterative solution for
LS there are also exact direct solvers, which have finite termination.  Direct solvers can be very efficient for sparse matrices
with good structure (small fill-in).  Also preconditioners can dramatically increase convergence of iterative methods -- especially
for near-chordal graphs.  So the statement seems somewhat too generic or misleading.

5) There is some relevant work that would be helpful comment on / cite:
There is a large body of work on multiplicative-update methods, e.g. see review by Satyan Kale, Elad Hazan and Sanjeev Arora,
"The Multiplicative Weights Update Method: A Meta-Algorithm and Applications", which have a variety of applications (including NMF,
online convex optimization, packing and covering).
For NMF specifically, there's work on separable (anchored) NMF, which has nice error guarantees, (Donoho et al, and Abhishek Kumar
and Vikas Sindhwani, "Fast Conical Hull Algorithms for Near-separable Non-negative Matrix Factorization".

6) For numerical experiments -- you compared to plain FISTA, but it's well known that practical convergence can often be dramatically improved with adaptive restarts, e.g. "Adaptive Restart for Accelerated Gradient Schemes", Donoghue and Emmanuel Candes.  Since you employed adaptive restarts in your method -- it seems only fair to use them for FISTA as well.   Furthermore, it'd be helpful to improve the numerical experiments -- run on more examples, and comment why it performs well on certain problems and less well on others (e.g. in terms of runtime). Why are FISTA and OA+DS are so poor on E2006train -- is this due to ill-conditioning / poor scaling?   Also mentioning timing results w.r.t. competing code is not always informative -- as e.g. using advanced compiler techniques (SIMD/AVX512, e.t.c.) vs. say python for-loops can make orders of magnitude runtime difference in itself.  (So reporting that fast NNLS [5] took 6.63 days may mean that the algorithm is poor, or that the implementation is poor, or both).

---

> ### Author Response · Authors · 2022-08-02
> **Response to Reviewer 5CSB**
>
> We thank the reviewer for their thoughtful comments and are encouraged by their positive feedback on our novelty, experiments, and clarity. We thank them for their reference  of Kumar-Sindhwani for NMF and have added it to our revised manuscript and commit to adding relevant literature on Poisson regression in the next revision.  We first answer the specific questions and then respond to the general comments.
>
> ### Why the natural residual is a good metric
>
> The natural residual may be viewed as a form of gradient mapping norm (commonly used in convex programming and in variational inequalities as a measure of convergence; see e.g., the textbooks [3, 4]). It has been used extensively in proving local error bounds for linear complementarity problems [2] and more generally in mathematical programming; see for example, the classical survey [5].
>
> ### Algorithm 2
>
> We apologize for the forward reference. Algorithm 2 is in the supplementary material (due to lack of space). This is indeed the one being referred to in Theorem 4.1. We have updated this in our manuscript.
>
> ### Batch size
>
> The question on the choice of batch sizes was raised by multiple reviewers; we, therefore, refer the reviewer to our top-level comment in this regard.
>
> ### Prevalence of NNLS+
>
> We refer the reviewer to our top-level response for examples of applications where NNLS+ makes sense.
> Among our datasets, real-sim and news20, are non-negative in the entries of the matrix, with possibly negative entries in the label vector: note that this is allowed in the NNLS+ formulation.
>
> Indeed, E2006train does have some negative entries, but as we state in our submission (Lines 296 - 297), our algorithm can (empirically) handle this case, and we, in fact, observe our algorithm to be the most competitive in this dataset, possibly due to its singularity (which SI-NNLS+ can handle well but neither OA+DS nor FISTA seem to).
>
> More broadly, we expect our algorithm to (empirically) converge well on datasets for NNLS; however, it is unclear to us if we can get rates (both theoretical and practical) as good as the ones we got (and proved) for NNLS+: specifically, the non-negativity condition is only used in the last step to connect the error bound to $f(x^*)$; eliminating the non-negativity prevents us from getting guarantees as strong as those in our main theorems (Theorem 3.5 and 4.1).
>
> ### Does NMF use NNLS+?
>
> To the best of our knowledge, existing NMF algorithms for many applications (see the ones listed in our top-level response) do in fact use NNLS as their subroutines, and the datasets NMF is applied to satisfy our assumed non-negativity. Therefore, developing a faster algorithm for NNLS+ makes sense for NMF. We have not tested against state-of-the-art NMF algorithms since we consider this to be beyond the scope of our paper.
>
> ### Exact direct solvers for linear regression
>
> We thank the reviewer for this insightful comment. We have clarified in the abstract that our statement is for the dimension-independent worst case oracle complexity. It is further true that classical methods such as variants of the Lawson-Hanson algorithm terminate in finite time on NNLS, although they are known to be quite slow in practice, at least on general instances. We have rephrased the introduction to clarify that our focus is on *standard* first-order methods, with dimension-independent iteration complexity. Finally, we believe that the good performance of our algorithm without any need for preconditioners is another strength of our work.
>
> ### Commentary on different performances across datasets
>
> We believe that the singularity of E2006train could possibly be the reason FISTA and OA+DS do not perform as well as SI-NNLS+.
>
> We thank the reviewer for their astute observation about using restart more fairly and have incorporated this in our revised manuscript.  We have already updated real-sim and news20 for using FISTA and OA+DS with restart. We are glad that SI-NNLS+ is still better than FISTA and OA+DS with restart on real-sim.
>
> We added an experiment of APCG [1] with restart, a variant of accelerated stochastic projected gradient descent, to real-sim and news20, against which SI-NNLS+ performs better or equivalent. We will add the experiments to E2006train in our revised manuscript.
>
> [1] Lin Q, Lu Z, Xiao L. An accelerated proximal coordinate gradient method. Advances in Neural Information Processing Systems, 2014, 27.
>
> [2] Mangasarian, O. L., & Ren, J. (1994). New improved error bounds for the linear complementarity problem. Mathematical Programming, 66(1), 241-255.
>
> [3] Facchinei, F., & Pang, J. S. (Eds.). (2003). Finite-dimensional variational inequalities and complementarity problems. New York, NY: Springer New York.
>
> [4] Nesterov, Y. Lectures on convex optimization. Vol. 137. Berlin: Springer International Publishing, 2018.
>
> [5] Pang JS. Error bounds in mathematical programming. Mathematical Programming. 1997 Oct;79(1):299-332.

---

> > ### Comment · Reviewer_5CSB · 2022-08-04
> > **Thanks!**
> >
> > Thank you for the detailed feedback, and clarifying all the points I raised!

---

### Official Review · Reviewer_mTbb · 2022-07-11

**Rating:** 6
**Confidence:** 3
**Soundness:** 3 good
**Presentation:** 3 good
**Contribution:** 2 fair

**Summary:**

The paper considers NNLS (nonnegative least squares) problems with an element-wise nonnegative data matrix and argues that in this scenario it is possible to obtain stronger guarantees than for traditional least squares. To support this claim, a scale-invariant iterative algorithm is provided that constructs an epsilon-multiplicative approximate solution. To obtain their approximation guarantees, the authors use the iterates of the algorithm to construct a sequence of upper estimates of the duality gap. The gap estimate itself is computed as a difference between an upper and lower bound on the Lagrangian. The gap estimate then enables them to obtain a bound on the primal error. The authors then describe how their proposed method can be combined with adaptive restart to obtain a linear convergence rate. The paper concludes with numerical experiments where the proposed method is compared to several competing optimization algorithms. The competing algorithms are outperformed in terms of runtime on one of three datasets while being competitive on the other two.


**Questions:**

It is a bit confusing that in the plots in Figure 1, there is no consistent choice of batch sizes even for the same dataset, e.g. for the new20 dataset, the batch size 50 is used in the plot for the number of data passes (e) but batch size 10 is used for the time plot (f). How were the batch sizes chosen for each plot? The effect of the choice of batch size could be further investigated in an experiment.



**Limitations:**

The paper does not contain any discussion of the societal impact of the paper.

**Strengths And Weaknesses:**

- Originality:
The paper provides a novel algorithm and proves its convergence. Moreover, the authors show how it can be adapted by combining it with a restart mechanism to achieve linear convergence. In the experiments it is shown to achieve competitive results.

- Quality:
The theoretical analysis seems to be sound but I was not able to check it in detail. The experimental evaluation is a bit weak and could be extended to more datasets.

- Clarity:
The structure of the paper could be improved, as the algorithm Lazy SI-NNLS+ is only defined in the appendix but already used in Theorem 4.1. It should at least be stated in the main paper that Lazy SI-NNLS+ is in fact equivalent to Algorithm 1, as proven in the Appendix.

- Significance:
The paper's contribution is mostly technical and allows the authors to obtain stronger theoretical guarantees compared to earlier work. The techniques used to obtain those improved theoretical guarantees (including the use of the adaptive restart technique) may be of value for the optimization and machine learning community. On the other hand, the experimental evaluation shows that the proposed algorithm is competitive but not consistently better than competing algorithms.


----
EDIT: Taking into account the feedback of the authors during the discussion phase, I have increased my score.

---

> ### Author Response · Authors · 2022-08-02
> **Response to Reviewer mTbb**
>
> We thank the reviewer for their thorough review: we are encouraged by their appreciation of our technical work and grateful to them for their suggestions on improving clarity and experiments.
>
> We have incorporated their suggestion on improving clarity of Lazy SI-NNLS+ (please see the blue text in our updated manuscript). We now address their major remarks and questions.
>
> ### Choice of data set
>
> The datasets we choose are large and the matrices singular (or nearly so), which is representative of many real-world, large-scale NNLS+ problems that we are concerned with in this paper.
>
> For example, in the dataset E2006train, the size of the matrix is $16087\times 150360$, and the number of nonzeros entries $19971015$; the spectral norm of the matrix is $5\times 10^{-6}$. The algorithms FISTA and OA+DS do not perform well on this dataset, which we suspect is due to the unfavorable singularity. The news20 and real-sim datasets have a spectral norm of roughly $10^{-3}$.
>
> We hope that our diverse choice of datasets effectively demonstrates that the convergence rate of SI-NNLS+ is affected only by the number of non-zero entries of the input dataset while being agnostic to the singularity of the problem.
>
> ### Comparison with other algorithms
>
> In the experiments on nonsingular dataset real-sim and news20, SI-NNLS+ is at least comparable to or better than OA+DS and FISTA. On news20, OA+DS is better in terms of time because the sparsity of the dataset can make use of the hardware to speed up the full update. On the other hand, for the singular data set E2006train, SI-NNLS+ is much better than the competitive algorithms. To further support our results, we added a comparison against APCG [1] with restart on real-sim and news20, and SI-NNLS+ performs better than APCG in terms of time on real-sim and equivalently in terms of time on news20. We will add experiments on APCG to E2006train in our revised manuscript.
>
> ### How the batch sizes were chosen
>
>  We thank the reviewer for pointing out the inconsistency in our batch sizes and have updated our experiments to fix this. We refer the reviewer to our top-level comment in reference to the choice of batch sizes (we answered it at the top since it was raised by multiple reviewers).
>
> [1]Lin Q, Lu Z, Xiao L. An accelerated proximal coordinate gradient method[J]. Advances in Neural Information Processing Systems, 2014, 27.

---

> > ### Comment · Reviewer_mTbb · 2022-08-08
> > **Thank you for your response.**
> >
> > Thank you for your response. I also appreciate the changes done to the clarity of the manuscript, the additional experimental comparison as well as the clarification on the choice of the batch size. Taking everything into account, I will therefore increase my score.

---

### Official Review · Reviewer_nhFt · 2022-07-13

**Rating:** 7
**Confidence:** 4
**Soundness:** 3 good
**Presentation:** 4 excellent
**Contribution:** 3 good

**Summary:**

This work studies Nonnegative (linear) least squares problems (NNLS) where the data itself is nonnegative. This work shows an interesting result that the nonnegativity of data makes the problem easier. In particular, they are able to present a nice and clean result of error bound (or complexity bound given an $\epsilon$ error) that is independent of any matrix constants (for example operator norm of the matrix), in this nonnegative coefficient/data case. With a new adaptive restart strategy, a linear convergence theory is also established. As the authors already mentioned, this is the first theoretical guarantee for NNLS+ (with nonnegative coefficient matrix $A$) that simultaneously satisfies the properties of being scale-invariant, accelerated, and linearly-convergent. Numerical experiments demonstrate the superior performance of the proposed method on large-scale data.

**Questions:**

1. The authors did a great job on improving the time complexity dependence from $O(1/\sqrt{\epsilon})$ to $O(\log(1/\epsilon))$ by introducing adaptive restart. I wonder if $O(\log(1/\epsilon))$ (linear convergence) the best one can do in theory, or, is superlinear convergence even possible in the NNLS/NNLS+ setting?

2. The running time of the proposed method seems to be quite sensitive to the batch size $bs$. Ideally, different $bs$ should be used for handling different sparsity levels of matrices. Is there a way to (roughly) choose this parameter before running the algorithm? It seems to be a complicated problem, as it also depends on how the hardware works.


**Limitations:**

Yes.

**Strengths And Weaknesses:**

Strengths:

1. This work has a very interesting and novel idea that uses the assumption of nonegativity of data to accelerate the algorithm and establish the improved theory.

2. The theory looks very nice and clean. Most importantly, the theoretical results look strong. In particular, the time complexity bounds for the proposed algorithm(s) are scale invariant. Indeed, they primarily rely on nnz($A$), and do not rely on its matrix norm. This nnz($A$) dependence also suggests that the theory and the proposed methods handle sparse matrices well (even though their dimension could be large).

3. The results from the numerical experiments look impressive.

4. This work is clearly motivated, well-organized, and very well-written. The proofs also look correct. I really enjoyed reading this paper.

Weaknesses:

Given the current state of the paper, the only "weakness" I can think of is its nonegative assumption of data, which may limit its applications. However, I would NOT say that it is really a weakness, since it is this nonegative assumption that brings advantages to the proposed methods (it also motivates the entire paper).

In summary, this work has made a strong contribution to the field of NNLS. It is well-written, and its idea is novel. Thus, I highly recommend accepting this manuscript.

---

> ### Author Response · Authors · 2022-08-02
> **Response to Reviewer nhFt**
>
> We thank the reviewer for their detailed review and interesting questions. We are deeply encouraged by their appreciation of our submission with regards to its novelty and elegance of ideas, strength of both the theory and experiments, as well as our presentation. We respond to the specific questions below.
>
> ### Is $\log(1/\epsilon)$ the best possible theoretical rate?
>
> We believe that for a quadratic program (of which our problem is an example), it may not be possible to obtain a superlinear rate of *global convergence*, but this may be possible for local convergence. It would be interesting to obtain a reference for this, but we have been unable to do so as of now. If we can find a reference, we would add it to our revised manuscript since that would either suggest optimality of our result or a clear open problem to work on.
> ### How do we choose the batch size
>
>  We refer the reviewer to our top-level comment (since this question was raised by multiple reviewers).
>
> ### Non-negative assumption of data
>
>  We agree with the reviewer that non-negativity lends itself to properties that help us get our accelerated, width-independent, and $\epsilon$-multiplicative result. We would like to additionally refer the reviewer to our top-level response wherein we justify, with ample examples, this assumption.
>
> We again thank the reviewer for their time and effort spent in reviewing our paper and for their questions. We hope all our responses, including the top-level one motivating our study of NNLS+, and our revised manuscript (with new text in blue), with most of the suggestions by reviewers incorporated in it, elevate their evaluation of our paper.

---

> > ### Comment · Reviewer_nhFt · 2022-08-08
> > **Response to the rebuttal**
> >
> > I thank the authors for the response. The comments by other reviewers and authors do not change my opinion that this is a solid paper. Therefore, I prefer not to change my score.

---

### Official Review · Reviewer_mKSw · 2022-07-17

**Rating:** 6
**Confidence:** 4
**Soundness:** 3 good
**Presentation:** 3 good
**Contribution:** 2 fair

**Summary:**

The paper studies the important non-negative least squares problem. The authors obtain stronger guarantees than for traditional least squares by assuming that the data matrix A is also non-negative. They showed that the problem can be solved up to multiplicative error and with complexity that is independent of any matrix constants.

**Questions:**

Line 29: "without the need to tune a regularization parameter or perform cross-validation [33, 6, 13]". I would add the references https://ieeexplore.ieee.org/document/8022909, https://ieeexplore.ieee.org/document/5608519 and https://arxiv.org/abs/1901.05727 given the theoretical importance that they have for the problem.

Line 32: "This is true, for example, in problems arising in image processing, computational genomics, functional MRI". It would be very interesting to have at least one example where this is clearly the case and/or references for each of these applications where it is crucial to have positive data since this is crucial for the paper.

Lina 40: Even though it seems to be correct, it is not well written. I would suggest rewriting this part.

Line 48: what is nnz(A)? Is it the number of nonzero elements of A? It should be clearly defined already at line 48.

Line 57: "What is significant about Theorem 1.1 is the invariance of the computational complexity to the scale of A—it does not depend on any matrix constants". Could you please explain the nature of this invariance? It is something that I didn't understand from the manuscript. Why does the positivity data give this invariance "for free" to the least squares problem? Would it be the case for other constraints as well?

Line 87: "for problems with strong convexity, a property not satisfied by underdetermined systems". If the matrix satisfies RIP, this is a form of strong convexity, at least when applied to sparse vectors. This sentence needs to be changed and more should be commented on that issue. I think that TNT-NN still works well for problems without strong convexity, even though there are no theoretical results to support it.

Line 100: "[16] and [24] mentioned in the preceding paragraph provide only asymptotic convergence guarantees". As far as I know, reference [24] (or its auxiliary paper https://ieeexplore.ieee.org/abstract/document/8425520) does not establish convergence of TNT for NNLS. It is a good algorithm in practice but without any theoretical guarantees.

Line 104-114: It would be simpler to visualize these algorithms on a table with the different results and assumptions that each one of them needs.

Line 142: I understand the change of variables. But matrix A will change as well. This should be stated in the paper. And what are the implications of this fact?

I dont understand the Lagrangian defined after line 158 and its construction is a crucial part for the analysis, if I understood correctly. I understand that you recover your primal problem from it but I don't understand why it is natural or what is the connection with the "standard" Lagrangian. Am I missing something obvious?

Line 182: Implementation version of Lazy SI-NNLS+. Is "Lazy" part of the name of the algorithm? If I understand correctly, in line 701, you described that the update of \tilde(x)_k and \tilde(y)_k would not be efficient if the matrix is sparse and, therefore, you have a lazy version of the algorithm. It would be interesting to briefly comment on this difference in the main text. I understand that on line 744 you mention the equivalence between Algorithms 1 and 2 but this should be in the main text somehow.

Line 227: You employ a randomized coordinate update in order to reduce the per-iteration complexity. But can you get a deterministic result as well without the randomized coordinate update?

Line 277: "Numerical experiments and discussion". I understand that there is a lack of space but for this kind of contribution I think it is crucial to have detailed numerical experiments. Or have more extensive experiments in the appendix. In the case of NNLS, a comparison with the Lawson-Hanson algorithm is very important given how much it is used in the literature. Maybe a comparison with TNT-NNLS as well since it was mentioned and it works really well in practice. I would also add a comparison with PGD and accelerated PGD with restart. I think that a comparison of all algorithms in situations where the matrix does not have RIP as well as matrices that are very ill-conditioned should be performed. And tests with synthetic and simple data to understand better what is going on. Also, differences between problems with sparse ground-truth versus dense ground truth. This is really important to convince a general audience of the usefulness of the algorithm in different situations.

Since the matrix A has all the entries positive, the matrix obviously satisfies the M_{+} condition (see, e.g., https://ieeexplore.ieee.org/document/5608519). What happens to your algorithm when the matrix A does not satisfy the M_{+} condition but does not have all the entries positive? What is the behaviour of the algorithm for the case that the ground truth is sparse?


**Limitations:**

As mentioned before, it is not clear how important the NNLS+ problem is and how the technique developed in this paper would be useful for NNLS problems under more general assumptions.

Also, it is not clear if the results are a simple application/modification of the VRPDA algorithm or not.

Moreover, the numerical section is quite limited and more extensive experiments must be performed.

**Strengths And Weaknesses:**

The paper is clear and develops a quite fast algorithm for the NNLS+ problem. It shows convergence by using a nice combination of a relaxed gap function combined with a random coordinate update.

The problem for me is how relevant the problem NNLS+ is. I think that the argumentation should be stronger because it is not clear, in principle, how and when this particular instance of NNLS would be interesting. Very precise references linking it to the applications should be presented. For example, Proposition 2.1 is very strong and it is not so clear what would happen without it (e.g. the choice of the box-constrained property of the optimum).
The numerical section needs to be reformulated, in my opinion, and more extensive (and simpler) numerical results should be presented, as I discussed in the "questions" section. Sparse vs dense ground truths usually present different behavior for the NNLS problem. What happens with SI-NNLS+in this case?

Even though something was mentioned on line 180, it would be important to stress, at least in the appendix, why this new algorithm is not a simple application/modification of the VRPDA algorithm in order to make the importance of novelty even more concrete.

---

> ### Author Response · Authors · 2022-08-02
> **Response to Reviewer mKSw (1/3)**
>
>
> We are deeply grateful to the reviewer for their detailed review of our submission, particularly their insightful questions about the significance of the non-negativity in our problem, the subtleties of scale-invariance, and our choice of experiments, all of which we answer shortly.
>
> We also greatly appreciate the large number of references they provided and have incorporated these, along with all suggestions for improving clarity (please see our updated manuscript with the new text in blue). We hope our edits improve our manuscript’s clarity and also better position our work in the landscape of existing results on this problem.
>
> We now address all the major questions and remarks. We have added most of these explanatory comments in the updated manuscript in blue text and will add the rest if accepted since the space constraints currently limit the amount of text we can add.
>
> ### Applications of non-negativity
>  As we state in the top-level comment addressed to all reviewers, we acknowledge the scant literature on NNLS+ per se. However, we clarify that our motivation to study NNLS+ (with the additional structure on $A$ in comparison with NNLS) **stems from our observations that in many prominent works on NNLS (such as on data arising in many medical applications like fMRI [2], EEG spectra [3], and pulse oximetry [7]), the experimental data used does indeed satisfy these structural assumptions** (we refer the reviewer to the top-level comments for more applications; we have included these references in our updated manuscript). One of our novel contributions, therefore, is in **recognizing this fact and using it to get faster rates (both theoretically and empirically) for this problem**.
>
> ### Novelty over VRPDA2
> Our algorithm differs both procedurally and technically from VRPDA2. Firstly, VRPDA2 uses the $\ell_2$-norm regularizer, and, in contrast, we use the special $\Lambda$-norm regularizer that depends on the problem matrix and confers useful properties on our algorithm. We also do not use a dual regularizer as in VRPDA2. Secondly, our analysis differs from VRPDA2 in that it makes a much sharper use of the problem structure and directly minimizes the primal-dual gap bound, as opposed to VRPDA2, which works with estimate sequences; this makes our algorithm more transparent and intuitive. (We have included this in our updated manuscript)
>
> ### The nature of the scale-invariance and role of non-negativity of $A$
> To set the stage to better explain this, we first recall that multiplicative approximation is defined by $f(x) - f(x^*) \leq \epsilon|f(x^*)|$ and that changing the scale of the objective does not change multiplicative approximation, as both sides are multiplied by the same constant. Further, our change of variable is *reversible*, which implies ${f}(\hat{x}) = \bar{f}(x)$, and this does not affect the approximation.
>
> With this background, we note that in the final bound,  $\| \hat{x}\_0 - \hat{x}^* \|\_{\mathbf{\Lambda}}^2$ is bounded by $|f(\hat{x}^*)| = |\bar{f}(x^*)|$ for $x\_0 = \mathbf{0},$ regardless of scale.
>
> The reason $\|A\|$ does not show up in the final bounds (thereby rendering our algorithm “scale-invariant”) is because Proposition 2.1 allows bounding $\| x_0 - x^* \|\_{\mathbf{\Lambda}}^2$ by $|f(x^*)|,$ where we crucially use the non-negativity of $A$ and $x.$ This does not seem possible for general $A.$ However, an additive (as opposed to multiplicative) error bound can be obtained even with the more general A with only small updates to the analysis. This bound would necessarily depend on the scale of $\mathbf{A}$. The choice of the regularizer $\frac{1}{2}\|\cdot-x_0\|_{\mathbf{\Lambda}}^2$ is also crucial here.
>
> We add that scale-invariance in general is a very important feature in problems with data matrices, since if there is a dependence on the width, then the algorithm is technically not polynomial-time. This feature has, in fact, been an object of extensive study in the long line of works on packing and covering linear programs [5, 8] and its variants such as a fair packing [6].
>
> ### Construction of Lagrangian
> The Lagrangian is constructed in a way that one can derive the primal problem from it while also localizing the matrix in the bilinear term and ensuring coordinate-wise separability of the terms involving either only the dual or only the primal variable since this greatly simplifies our analysis.

---

> > ### Author Response · Authors · 2022-08-02
> > **Continued Response to Reviewer mKSw (2/3)**
> >
> > ### Deterministic algorithm?
> >  As noted by the reviewer, the randomness was introduced to reduce the per-iteration cost; we believe that for our analysis to go through, the use of this randomness is critical since we cannot afford to pay the cost of updating an entire vector in each iteration and need to choose one coordinate. In particular, if we were to resort to a deterministic algorithm with full vector updates, we would need to pay an additional either $\sqrt{n}$ or matrix-dependent factor in the convergence bound, which would be incurred when bounding the error terms of the form $\langle\mathbf{y}\_k - \mathbf{y}\_{k-1}, \mathbf{A}(\mathbf{x}\_k - \mathbf{x}\_{k-1})\rangle.$
> >
> > ### Experiments: updated per reviewer’s remarks
> > Per the reviewer’s suggestion, we have now added to our experiments a comparison against Accelerated Proximal Coordinate Gradient (APCG) [1] (a variant of accelerated stochastic projected gradient descent) on the dataset real-sim and news20. We find that SI-NNLS+ is better than APCG in terms of time and comparable in terms of the number of data passes on real-sim.
> >
> > We will implement APCG as well for E2006train in the updated manuscript.
> >
> > In reference to the suggestion to use accelerated PGD, we note that in our problem, FISTA is accelerated projected gradient descent.
> >
> > We updated the experiment for FISTA and OA+DS with restart for real-sim and news20, and this improved FISTA and OA+DS, but still, we are happy to report that SI-NNLS+ (bs=1) is better than these two algorithms for real-sim. We will update corresponding results for E2006train in the next revision of our manuscript.
> >
> > The reason we do not compare against Lawson-Hanson and TNT in our experiments is because they are not first-order algorithms, as a consequence of which they require a significant amount of memory to run, leading to “out of memory” errors for even moderately-sized data matrices. We have corrected the sentence that incorrectly claims a theoretical convergence guarantee on TNT by [4] — we thank the reviewer for this correction.
> >
> > We commit to finding more diverse datasets as suggested by the reviewer (such as ones without RIP, with ill-conditioning, simple, synthetic, etc.) in the next revision of our manuscript.
> >
> > ### $M_+$ condition
> > We thank the reviewer for pointing us to this paper (and this interesting property, which could be viewed as a generalization of element-wise non-negativity, as the reviewer alludes to). To answer their question, in problems with the $M_+$ property but without element-wise non-negativity, we believe our algorithm will still converge, but the guarantees would not be as strong as those we have obtained for our case. Obtaining comparable guarantees would likely be non-trivial and certainly a highly interesting future direction for this work.
> >
> > ### Problems with sparse ground truth
> >  We thank the reviewer for this insightful question. In the sparse case, Theorem 3.5 may be interpreted as $\mathbb{E}[f(x_K) - f(x^*)] \leq \varepsilon \min\\{\mathrm{nnz}(x^*),|f(x^*)|\\}$. However, it may require new ideas to get a result with restarts similar to that in Theorem 4.1 since the monotonicity of $\mathrm{nnz}(x_K)$ would need to be maintained for restart to work. We do not have a data set with sparse ground truth solution and would be happy to run experiments on one if the reviewer could provide a pointer to any such dataset.
> >
> > We again thank the reviewer for their time and effort put into such a thoughtful and constructive review. We hope our responses, particularly the ones motivating NNLS+, along with our revised manuscript incorporating most of their suggestions, elevate their view of our manuscript.

---

> > > ### Author Response · Authors · 2022-08-02
> > > **Continued response to Reviewer mKSw (3/3)**
> > >
> > > [1] Lin Q, Lu Z, Xiao L. An accelerated proximal coordinate gradient method[J]. Advances in Neural Information Processing Systems, 2014, 27.
> > >
> > > [2] Andersen, A. H., & Rayens, W. S. (2004). Structure-seeking multilinear methods for the analysis of fMRI data. NeuroImage, 22(2), 728-739.
> > >
> > > [3] Martínez-Montes, E., Sánchez-Bornot, J. M., & Valdés-Sosa, P. A. (2008). Penalized PARAFAC analysis of spontaneous EEG recordings. Statistica Sinica, 1449-1464.
> > >
> > > [4] Myre, J. M., Frahm, E., Lilja, D. J., & Saar, M. O. (2017). TNT-NN: a fast active set method for solving large non-negative least squares problems. Procedia Computer Science, 108, 755-764.
> > >
> > > [5] Allen-Zhu, Z., & Orecchia, L. (2019). Nearly linear-time packing and covering LP solvers. Mathematical Programming, 175(1), 307-353.
> > >
> > > [6] Diakonikolas, J., Fazel, M., & Orecchia, L. (2020). Fair packing and covering on a relative scale. SIAM Journal on Optimization, 30(4), 3284-3314.
> > >
> > > [7] Wukitsch, M. W., Petterson, M. T., Tobler, D. R., & Pologe, J. A. (1988). Pulse oximetry: analysis of theory, technology, and practice. Journal of clinical monitoring, 4(4), 290-301.
> > >
> > > [8] Wang, Di. Fast Approximation Algorithms for Positive Linear Programs. University of California, Berkeley, 2017.

---

> > > > ### Comment · Reviewer_mKSw · 2022-08-08
> > > > **Thank you for your detailed comments**
> > > >
> > > > Thank you very much for the top-level response as well as the specific comments and for addressing some of the issues that I've mentioned. I increased my score based on the explanations and the changes provided in the paper. In particular, thanks for clarifying the novelty over VRPDA2 and the nature of the scale-invariance. That being said, I would like to ask the authors if a github page will be available with the code and the datasets. It would be very important to have a reproducible paper.
> > > >
> > > > Also, I still would like to add the following comments
> > > >
> > > > 1 - It seems very strange to me that FISTA with restart does not perform well in all the examples shown in the paper, if I understood correctly. That is why it would be nice to test these algorithms with synthetic data where we can perfectly control what happening in terms of condition number, sparsity, dimensions, etc. I would like to numerically test a few things if the authors make the code available.
> > > >
> > > > 2 - I would strongly emphasize, before the proof of Proposition 2.1, that it strongly needs the non-negativity of A and x. It is fair to say that it does not hold in general and you need it for your result. I would add the explanation that you provided here in the supplementary material.
> > > >
> > > > 3 - For each of the three experiments, I would add information such as the spectral norm and condition number of the matrix A.
> > > >
> > > > Thank you again for all the work and the response.

---

> > > > > ### Author Response · Authors · 2022-08-09
> > > > > **Thank you!**
> > > > >
> > > > > Thank you very much for considering our response and increasing your score; we very much appreciate it.
> > > > >
> > > > > We do plan to make the code publicly available on github once we have had the chance to clean it up after the added code from the rebuttals. If you have suggestions for generating synthetic data on which it would be helpful to run more experiments or any pointers to similar experiments in the literature), please do share those with us and we'll be happy to include them in the final version (though, due to page constraints, those will likely end up in the appendix).
> > > > >
> > > > > That's a good point about emphasizing that non-negativity is needed for Proposition 3.1 and including the values of the spectral norm and the condition number for A -- we will add those.
> > > > >
> > > > > Thank you again for all the constructive feedback.

---

### Author Response · Authors · 2022-08-02
**Top-level response to all reviewers (1/3)**

(1/3)

We are deeply grateful to all the reviewers for well-thought out reviews of our submission, their appreciation of our technical ideas, and their multiple constructive suggestions in helping us strengthen our work.

We will address all their major questions and remarks by responding directly to each review, but here at a top level, we first briefly reiterate our two key contributions.

We have also added the requested additional explanatory comments and references in our revised manuscript (please see the updated manuscript with the new content in blue text).

### Motivation for the problem we study

We initiate a systematic study of the algorithmic complexity of the problem NNLS+ (non-negative least squares with the non-negative constraint matrix as well as the problem variable constraint). A common theme among the reviews seems to be the lack of prevalence of NNLS+ in comparison with NNLS. While we acknowledge the scant literature on NNLS+ per se, **we justify our choice to study NNLS+ on the basis of our observation that, in many of the papers that study NNLS, the (real-world) datasets experimented on do, in fact, satisfy the (stronger) assumptions we impose in NNLS+**.

Some prominent practical and theoretical applications where the above property appears include:
- in the calibration and evaluation of pulse oximeters [11, 12], a device used to measure the oxygen saturation level in the blood and therefore of critical importance in emergency rooms in hospitals and, more recently, during the COVID-19 pandemic to monitor patients’ blood oxygen levels;
- in chemistry: for example, in [3] wherein the datasets comprise fluorescence spectroscopic measurements, which are non-negative;
in image processing [15] and also with specific focus on MRI data [19], functional MRI (fMRI) [9, 13] and EEG spectra [10] data;
in computational genomics [16, 17];
- In statistical procedures in observational astronomy to, for example, “calibrate and quantify the cosmic distance scale necessary for the study of large-scale structure of the universe [14]”.
- in non-negative matrix factorization [1]: Quoting from the textbook [2], “...*Many real-world data are nonnegative and the corresponding hidden components have a physical meaning only when nonnegative. In practice, both nonnegative and sparse decompositions of data are often either desirable or necessary when the underlying components have a physical interpretation*…” (c.f. Lines 22 - 24 of our submission). Several examples of such datasets may be found in [2, 8], with two commonly used ones (for example, in [4] to which we compare our results) being news20 and real-sim and [7] including examples for text mining and spectral data analysis.

We hope that these important examples sufficiently demonstrate the ubiquity of the applicability of NNLS+. The novel properties of the optimal point we deduce from these structural assumptions (c.f. Proposition 2.1) let us obtain far improved results than those for NNLS, reiterated next.

We continue our response in the comment below.

---

> ### Author Response · Authors · 2022-08-02
> **Continued top-level response (2/3)**
>
> ### Our results
>
> Using our insights on the problem structure atop a primal-dual algorithm inspired by VRPDA2, we obtain **the first accelerated, scale-invariant, $\epsilon$-multiplicative approximate algorithm for NNLS+**. Each of these features is considered important (and, in general, difficult to achieve) in its own right in optimization [5, 6], and we, therefore, consider this result in itself a highly non-trivial contribution to a commonly encountered practical problem. Additionally, we combine this result with the adaptive restart strategy in a novel way to obtain an additional **linear convergence guarantee** and supplement all our theoretical results with experiments on real-world data.
>
> Aside from the importance of these results on a highly practically relevant problem, we believe (and hope) that our specific techniques may be extended to many important problems where acceleration and width-indepence are much sought after, for example [5, 6].
>
> ### Some comments on the batch size
>
> From a theoretical perspective, large batch sizes generally do not help: if we used larger batch sizes, in our model, we would be paying an additional $\sqrt{s}$ factor in the theoretical runtime, where $s$ is the batch size. This comes from bounding the error terms of the form $(y_k - y_{k-1})^\top (A(x_k - x_{k-1}))$ in our analysis.
>
> On the other hand, the reason that larger batch sizes sometimes help in practice comes from the way that modern processors work. This is what the “instruction-level parallelism” refers to: modern processors can effectively parallelize computations with larger batches, thus reducing per-iteration cost.
>
> However, theoretically, because this is hardware-dependent, we do not have a good way of modeling this behavior to be able to incorporate it in our analysis
>
> In general, the problem of rigorously deducing the batch size is quite difficult and seems to depend on the problem size, regularity of the matrices involved, and the computer's computational power. As of now we are guided by heuristics in this task: in our case, simply choosing between the batch sizes of 1, 10, and 50 did the job for us. We expect batch sizes between 10 and 100 to suffice. This is also in agreement with similar observations made in the paper [18].
>
> References for the papers cited in this top-level response are provided below.

---

> > ### Author Response · Authors · 2022-08-02
> > **Continued top-level response (3/3)**
> >
> >
> >
> > [1] Kim, Dongmin, Suvrit Sra, and Inderjit S. Dhillon. "Fast Newton-type methods for the least squares nonnegative matrix approximation problem." In Proceedings of the 2007 SIAM international conference on data mining, pp. 343-354. Society for Industrial and Applied Mathematics, 2007.
> >
> > [2] Cichocki, A., Zdunek, R., Phan, A. H., & Amari, S. I. (2009). Nonnegative matrix and tensor factorizations: applications to exploratory multi-way data analysis and blind source separation. John Wiley & Sons.
> >
> > [3] Bro, R., & De Jong, S. (1997). A fast non‐negativity‐constrained least squares algorithm. Journal of Chemometrics: A Journal of the Chemometrics Society, 11(5), 393-401.
> >
> > [4] Kim, D., Sra, S., & Dhillon, I. S. (2013). A non-monotonic method for large-scale non-negative least squares. Optimization Methods and Software, 28(5), 1012-1039.
> >
> > [5] Wang, Di. Fast Approximation Algorithms for Positive Linear Programs. University of California, Berkeley, 2017.
> >
> > [6] Nesterov, Y. (2018). Lectures on convex optimization (Vol. 137). Berlin: Springer International Publishing.
> >
> > [7] Berry, M. W., Browne, M., Langville, A. N., Pauca, V. P., & Plemmons, R. J. (2007). Algorithms and applications for approximate nonnegative matrix factorization. Computational statistics & data analysis, 52(1), 155-173.
> >
> > [8] Sra, S., & Dhillon, I. (2005). Generalized nonnegative matrix approximations with Bregman divergences. Advances in neural information processing systems, 18.
> >
> > [9] Andersen, A. H., & Rayens, W. S. (2004). Structure-seeking multilinear methods for the analysis of fMRI data. NeuroImage, 22(2), 728-739.
> >
> > [10] Martínez-Montes, E., Sánchez-Bornot, J. M., & Valdés-Sosa, P. A. (2008). Penalized PARAFAC analysis of spontaneous EEG recordings. Statistica Sinica, 1449-1464.
> >
> > [11] Jennis, M. S., & Peabody, J. L. (1987). Pulse oximetry: an alternative method for the assessment of oxygenation in newborn infants. Pediatrics, 79(4), 524-528.
> >
> > [12] Wukitsch, M. W., Petterson, M. T., Tobler, D. R., & Pologe, J. A. (1988). Pulse oximetry: analysis of theory, technology, and practice. Journal of clinical monitoring, 4(4), 290-301.
> >
> > [13] Jackson, E. C., Hughes, J. A., & Daley, M. (2018, May). On the generalizability of linear and non-linear region of interest-based multivariate regression models for fmri data. In 2018 IEEE Conference on Computational Intelligence in Bioinformatics and Computational Biology (CIBCB) (pp. 1-8).
> >
> > [14] Isobe, T., Feigelson, E. D., Akritas, M. G., & Babu, G. J. (1990). Linear regression in astronomy. The astrophysical journal, 364, 104-113.
> >
> > [15]  Esser E, Lou Y, Xin J. A method for finding structured sparse solutions to nonnegative least squares problems with applications[J]. SIAM Journal on Imaging Sciences, 2013, 6(4): 2010-2046
> >
> > [16] Li Y, Ngom A. Nonnegative least-squares methods for the classification of high-dimensional biological data[J]. IEEE/ACM transactions on computational biology and bioinformatics, 2013, 10(2): 447-456.
> >
> > [17] Silva G G Z, Cuevas D A, Dutilh B E, et al. FOCUS: an alignment-free model to identify organisms in metagenomes using non-negative least squares[J]. PeerJ, 2014, 2: e425.
> >
> > [18] Song, C., Lin, C. Y., Wright, S. J., & Diakonikolas, J. (2021). Coordinate linear variance reduction for generalized linear programming. arXiv preprint arXiv:2111.01842.
> >
> > [19]  Marchand AJ, Hitti E, Monge F, et al. MRI quantification of diffusion and perfusion in bone marrow by intravoxel incoherent motion (IVIM) and non-negative least square (NNLS) analysis. Magn Reson Imaging. 2014;32(9):1091-1096.

---

### Meta-Review · Area_Chair_BGA1 · 2022-08-25

**Recommendation:** Accept
**Confidence:** Certain

**Metareview:**

The reviewers generally agreed that the paper has novel and solid contributions. Moreover, they were satisfied with the authors' responses. Please incorporate the reviewers' suggestions in your revision.

**Award:**

No

---

### Decision · Program_Chairs · 2022-09-14

Accept